# Microwave and submillimeter wave scattering of oriented ice particles

Manfred Brath[1], Robin Ekelund[2], Patrick Eriksson[2], Oliver Lemke[1], and Stefan A. Buehler[1]

[1]Universität Hamburg, Faculty of Mathematics, Informatics and Natural Sciences, Department of Earth Sciences, Meteorological Institute, Hamburg, Germany
[2]Department of Space, Earth and Environment, Chalmers University of Technology, Gothenburg, Sweden

**Correspondence:** Manfred Brath
(manfred.brath@uni-hamburg.de)

**Abstract.** Microwave ($1\,\mathrm{GHz}$–$300\,\mathrm{GHz}$) dual-polarization measurements above $100\,\mathrm{GHz}$ are so far sparse, but they consistently show polarized scattering signals of ice clouds. Existing scattering databases of realistically shaped ice crystals for microwaves and submillimeter waves ($> 300\,\mathrm{GHz}$) typically assume total random orientation, which cannot explain the polarized signals. Conceptual models show that the polarization signals are caused by oriented ice particles. Only few works that consider
oriented ice crystals exist, but they are limited to microwaves only. Assuming azimuthally randomly oriented ice particles with a fixed but arbitrary tilt angle, we produced scattering data for two particle habits ($51$ hexagonal plates and $18$ plate aggregates), $35$ frequencies between $1\,\mathrm{GHz}$ and $864\,\mathrm{GHz}$, and $3$ temperatures ($190\,\mathrm{K}$, $230\,\mathrm{K}$ and $270\,\mathrm{K}$). The scattering data of azimuthally randomly oriented particles depends in general on the incidence angle and two scattering angles in contrast to total random orientation, which depends on a single angle. The additional tilt angle further increases the complexity. The simulations
are based on the discrete dipole approximation in combination with a self-developed orientation averaging approach. The scattering data is publicly available from Zenodo (https://doi.org/10.5281/zenodo.3463003). This effort is also an essential part of preparing for the upcoming Ice Cloud Imager (ICI) that will perform polarized observations at $243\,\mathrm{GHz}$ and $664\,\mathrm{GHz}$. Using our scattering data radiative transfer simulations with two liquid hydrometeor species and four frozen hydrometeor species of polarized GMI (GPM (Global Precipitation Measurement) Microwave Imager) observations at $166\,\mathrm{GHz}$ were conducted.
The simulations recreate the observed polarization patterns. For slightly fluttering snow and ice particles, the simulations show polarization differences up to $11\,\mathrm{K}$ using plate aggregates for snow, hexagonal plates for cloud ice and totally randomly oriented particles for the remaining species. Simulations using strongly fluttering hexagonal plates for snow and ice show similar polarization signals. Orientation, shape and the hydrometeor composition affect the polarization. Ignoring orientation can cause a negative bias for vertically polarized observations and a positive bias for horizontally polarized observations.

## 1 Introduction

Passive microwave (MW) observations are nowadays a standard tool for cloud observation. The ice cloud related sounding channels of passive microwave sensors typically do not possess a fixed polarization or they measure only at one polarization. Observations of polarization in view of MW and submillimeter (SubMM) remote sensing of ice clouds are still rare. Existing

passive microwave sensors that measure polarization are typically confined to frequencies below $100\,$GHz. Due to the low frequency, their sensitivity considering ice clouds is low (Buehler et al., 2007), though there still can be enough sensitivity for precipitating ice. However, these sensors are affected by surface contamination.

Currently, GMI (GPM (Global Precipitation Measurement) Microwave Imager, Hou et al., 2013) is the only spaceborne microwave radiometer that measures polarization above $100\,$GHz. In the past, MADRAS (Microwave Analysis and Detection of Rain and Atmospheric Structure, Defer et al., 2014) on board of Megha-Tropique also observed polarization at ice cloud related frequencies, but due to mechanical failure only till January 2013 (Shivakumar and Pircher, 2013). GMI and MADRAS observe polarization around $160\,$GHz. Defer et al. (2014); Gong and Wu (2017) and Zeng et al. (2019) showed MW observations of polarized scattering signals of clouds using GMI and MADRAS. Based on radiative transfer simulations, Defer et al. (2014) and Gong and Wu (2017) explained these polarized signals by the asphericity and a preferred orientation of the ice particles. Therefore, exploiting polarization can deliver additional information about the particle shape and orientation. Ice crystals have several shapes and sizes in reality. Furthermore, even the cases that have been explained by horizontally aligned particles consist in reality not only of particles limited to one orientation, but of particles with several different orientations, from which some may be more probable. With the upcoming ICI (Ice Cloud Imager, Eriksson et al. (2020); Bergadá et al. (2016); Buehler et al. (2012, 2007)) there will be additional polarized observations at $243\,$GHz and at $664\,$GHz. These polarized observations will deliver new insights about clouds and their structure, because of their higher sensitivity to ice clouds compared to GMI. The scattering data directly affects simulations and inversions of MW and SubMM ice cloud observations, as the scattering data describes the interaction between ice particles and the electromagnetic radiation. This limits the phenomena that can be considered and the amount of information that can be retrieved from the observations. Therefore, in order to exploit polarization, data of the scattering properties of oriented and realistically shaped particles is required.

Existing single scattering databases of realistically shaped ice particles for the microwave and submillimeter range, like the ones of Eriksson et al. (2018), Liu (2008) or Hong et al. (2009), assume total random orientation of the scatterers. This is often a reasonable assumption, but cannot explain polarized cloud signals. This requires oriented scatterers. The studies of Lu et al. (2016) and of Adams and Bettenhausen (2012) take orientation into account but are limited to frequencies below $94\,$GHz and $166\,$GHz, respectively.

This paper aims to simulate the MW and SubMM scattering data of realistically shaped ice crystals that possess arbitrary fixed orientations relative to the zenith direction under the assumption that there is no preferred orientation in azimuth direction. In reality ice crystals have myriads of shapes as shown for example by Libbrecht (2005) or Heymsfield et al. (2002). Here we consider only two types of ice crystals: hexagonal plates and aggregates consisting of hexagonal plates. The resulting single scattering database is publicly available from Zenodo (https://doi.org/10.5281/zenodo.3463003). The idea behind the scattering database is that the users can use the scattering data of a desired zenith orientation or combine the data of different zenith orientations to mimic any desired distribution of zenith orientations. The scattering database is structured so that it can be used together with the scattering database of Eriksson et al. (2018).

To simulate the scattering properties, the scattering of ice crystal from various incidence directions is simulated and consequently used to calculate orientation averaged scattering. Similar to the work of Eriksson et al. (2018), Adams and Bettenhausen

(2012), Hong et al. (2009) or Liu (2008) the scattering is simulated on the basis of the discrete dipole approximation (DDA,
Draine and Flatau (1994)). Furthermore, the simulated scattering properties of ice particles are used for radiative transfer
simulations of cloudy scenes to investigate their influence on actual brightness temperature observations.

The text is structured as follows: in Sect. 2 we explain the particle orientation. Sect. 3 provides an overview of the basic setup
and the simulated particles. Sect. 4 explains the scattering simulation. Sect. 5 shows some example results. Sect. 6 considers the
influence of the simulated scattering properties in view of radiative transfer simulations. In Sect. 7 we summarize the results.

## 2  Particle orientation

Particle orientation refers to how the main axes of the particle are oriented with respect to the local horizon and the azimuthal
reference. If the particle possesses a spherical symmetry, there is no particle orientation, because it does not matter from which
side the particle is viewed or how it is rotated - it will always look the same. The particles considered in this paper are not
spherically symmetric and therefore can be oriented.

In general, the orientation of a particle in a three-dimensional space can be described by a set of three parameters. There is
no unique set of these parameters. Depending on the definition of the rotation axes, there are different sets of these parameters.
The three Euler angles are one such parameter set. The Euler angles define the orientation of the particle (coordinate) system
relative to a fixed coordinate system, hereafter called laboratory system. The particle system is the coordinate system that is
attached to the particle. This means that if a particle is rotated, the particle system is rotated the same way. The laboratory
system stays under the rotation of the particle, whereas the particle system changes its orientation. The laboratory system and
particle system share the same origin. In this study, the Euler angles, which are shown in Fig. 1, are used according to the $zyz'$-
notation. The particle is first rotated by angle $\alpha$ around the laboratory Z-axis, then the particle is rotated by angle $\beta$ around the
particle Y-axis ($y'$) and lastly the particle is rotated by angle $\gamma$ around the particle Z-axis. The value ranges of the angles are

$$\alpha \in [0, 2\pi]$$
$$\beta \in [0, \pi] \tag{1}$$
$$\gamma \in [0, 2\pi]$$

These rotations are described by three orthogonal rotation matrices, see Appendix B for details. It is important to note that in
general the order of the rotations must not be changed, because the combination of rotations is generally not commutative.

In addition to the Euler angles, the orientation of the non-rotated particle is needed. As there is no absolute coordinate
system, the orientation of the non-rotated particle is in general arbitrary. Therefore, we define that the non-rotated particle lies
with its center of gravity at the origin of the laboratory system and all particle rotations are be relative to the origin of the
laboratory system. Furthermore, the non-rotated particle is defined to have its principal moments of inertia axes aligned along
the Cartesian coordinate axes, with the maximum inertia axis along the z-axis and the smallest along the x-axis (see Appendix
A). This means for a plate-like particle that its longest dimensions lay parallel to the x-y-plane. This is the orientation that one
intuitively expects for a falling plate-like particle in air. In reality the orientation of a particle determined by the interaction of

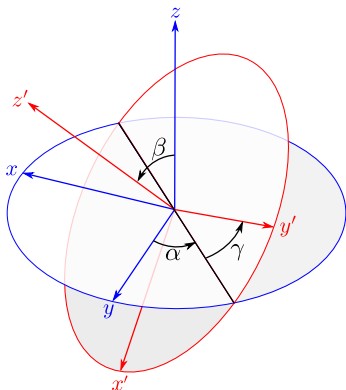

**Figure 1.** Euler angles

the gravitational force on one side and the drag force and other forces like e.g. electrical force on the other side (Khvorostyanov

and Curry, 2014). The drag force is determined by the interaction of particle and the surrounding air. Estimating the drag force

is a challenging task, as one has to solve the Navier-Stokes equations for that. Klett (1995) modeled the orientation of falling

ice crystals. Under turbulent free conditions falling plates with diameters $> 40\,\mu\text{m}$ and columns with lengths $> 30\,\mu\text{m}$ are on

average horizontally oriented. As most of the particles considered in our study are greater than $40\,\mu\text{m}$, we expect our definition

for the non-rotated particle to be reasonable. Though we do not consider column-like particles in the study, the study of Klett

(1995) suggests that even for them our definition is reasonable.

Within this study, we are not interested in the scattering of a single oriented particle but in the scattering of an ensemble of

particles that are oriented differently but otherwise are identical. Generally, the scattering properties of such an ensemble of

oriented particles are described by averaging the single scattering properties over the three Euler angles such that for example

for the scattering matrix $\boldsymbol{Z}_{\text{eo}}$ and the extinction matrix $\boldsymbol{K}_{eo}$ of an ensemble of orientated particles hold

$$\boldsymbol{Z}_{\text{eo}}\left(\theta_{inc},\phi_{inc},\theta_s,\phi_s\right) = \int\limits_0^{2\pi}\int\limits_0^{\pi}\int\limits_0^{2\pi} p_\alpha(\alpha)\,p_\beta(\beta)\,p_\gamma(\gamma)\,\boldsymbol{Z}\left(\theta_{inc},\phi_{inc},\theta_s,\phi_s,\alpha,\beta,\gamma\right)\,\mathrm{d}\alpha\,\mathrm{d}\beta\,\mathrm{d}\gamma \tag{2}$$

$$\boldsymbol{K}_{eo} = \left(\theta_{inc},\phi_{inc}\right) = \int\limits_0^{2\pi}\int\limits_0^{\pi}\int\limits_0^{2\pi} p_\alpha(\alpha)\,p_\beta(\beta)\,p_\gamma(\gamma)\,\boldsymbol{K}\left(\theta_{inc},\phi_{inc},\alpha,\beta,\gamma\right)\,\mathrm{d}\alpha\,\mathrm{d}\beta\,\mathrm{d}\gamma \tag{3}$$

with $\theta_{inc}$ the incidence polar angle, $\phi_{inc}$ the incidence azimuth angle, $\theta_s$ the scattering polar angle and $\phi_s$ the scattering azimuth

angle. $p_j\left(x\right)$ are probability density functions describing the distribution of particle orientation. Equations 2 and 3 implicitly

assume independent scattering, which is typically assumed in context of atmospheric radiative transfer. This means that the

scatterers are separated enough in distance so that their scattered waves do not interact and that there are no systematic phase

relations between the scattered waves (Mishchenko et al., 2000). In other words, Eq. 2 and 3 assume incoherent scattering.

We distinguish between two basic states of particle orientation

1. total random orientation (TRO)

2. azimuthal random orientation (ARO).

Both orientation states are explained in the two following subsections.

## 2.1 Total random orientation

Totally randomly oriented particles are defined as the orientation average over the three Euler angles in which the Euler angles are uniformly distributed. That is,

$$p_\alpha(\alpha) = p_\gamma(\gamma) = \frac{1}{2\pi} \tag{4}$$

$$p_\beta(\beta) = \frac{\sin\beta}{2} \tag{5}$$

(Mishchenko and Yurkin, 2017). Due to this averaging, totally randomly oriented particles have effectively a spherical symmetry. This implies that the scattering matrix of totally randomly oriented particles depends only, like the scattering matrix of spheres, on the scattering angle $\Theta$ i.e.,

$$\boldsymbol{Z}_{\mathrm{tro}}(\Theta) = \boldsymbol{Z}_{\mathrm{tro}}(\theta_{inc}, \phi_{inc}, \theta_s, \phi_s), \tag{6}$$

and $\boldsymbol{K}_{\mathrm{tro}}$ will have no angular dependency. The scattering angle $\Theta$

$$\cos\Theta = \cos\theta_{inc}\cos\theta_s + \sin\theta_{inc}\sin\theta_s\cos(\phi_s - \phi_{inc}) \tag{7}$$

is the angle between the incidence and the outgoing direction (see also Fig. D1). Eriksson et al. (2018), Ding et al. (2017), Liu (2008) and Hong et al. (2009) assume total random orientation in their databases.

## 2.2 Azimuthal random orientation

Azimuthally randomly oriented particles with a specific orientation to the horizon, also referred to as tilt or canting, are defined as the orientation average over $\alpha$ and $\gamma$ in which $\alpha$ and $\gamma$ are uniformly distributed as for total random orientation. The scattering matrix $\boldsymbol{Z}_{\mathrm{aro}}$ and the extinction matrix $\boldsymbol{K}_{aro}$ of azimuthally randomly oriented particles are thus calculated as

$$\boldsymbol{Z}_{\mathrm{aro}}(\theta_{inc}, \theta_s, \Delta\phi, \beta) = \int_0^{2\pi}\int_0^{2\pi} p_\alpha(\alpha)\, p_\gamma(\gamma)\, \boldsymbol{Z}(\theta_{inc}, \phi_{inc}, \theta_s, \phi_s, \alpha, \beta, \gamma)\, \mathrm{d}\alpha\, \mathrm{d}\gamma \tag{8}$$


$$\boldsymbol{K}_{\mathrm{aro}}(\theta_{inc}, \beta) = \int_0^{2\pi}\int_0^{2\pi} p_\alpha(\alpha)\, p_\gamma(\gamma)\, \boldsymbol{K}(\theta_{inc}, \phi_{inc}, \alpha, \beta, \gamma)\, \mathrm{d}\alpha\, \mathrm{d}\gamma. \tag{9}$$

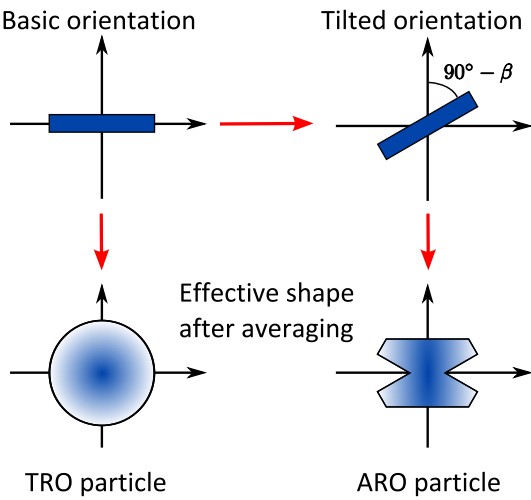

**Figure 2.** Schematic of the difference between totally random (TRO) and azimuthally random orientation (ARO) for columnar particle .

The averaging over $\alpha$ and $\gamma$ results in a rotational symmetry of the scattering matrix to the laboratory Z-axis (cylindrical symmetry). The orientation average results in an effective particle shape as indicated in Fig. 2. To get a better picture of it, assume that the particle rotates very fast around the laboratory Z-axis and the particle Z-axis to symbolize the orientation averaging. By rotation it creates an effective solid of revolution. Changing the tilt angle $\beta$ results in a different shape of this effective solid of revolution. Due to the cylindrical symmetry after orientation averaging, the averaged scattering matrix depends in azimuth only on the difference between incident and scattered azimuth direction. Whereas the scattering matrix of totally randomly oriented particles depends only on the scattering angle $\Theta$, the scattering matrix of azimuthally randomly oriented particles depends on the incidence polar angle $\theta_{inc}$, the scattering polar angle $\theta_s$, the difference of the incidence and scattering azimuth angles $\Delta\phi = \phi_{inc} - \phi_s$ and the tilt angle $\beta$. Without any loss of generality, the azimuth incidence angle $\phi_{inc}$ is set to $0°$ for the azimuthally randomly oriented case from here on. It is important to note that the azimuthal symmetry does not mean that the scattering matrix $\boldsymbol{Z}_{\mathrm{aro}}$ is symmetric. This depends on the symmetry properties of the particles and the orientation of the rotation axes relative to the symmetry axes. To get a better idea of it, assume a flag rotates fast around its flagpole in counterclockwise direction as shown in Fig. 3. The flag has a red frontside, a blue backside and its hoist is to the left. Independent from which side we look on the flagpole, the projections of the red front side are always seen on the right side of the flagpole and the projections of the blue backside are always seen on the left side. If both sides of the flag have the same color, then the projections on both sides will look the same. Although the rotation results in a rotational symmetry around the flagpole, the actual image we see depends on the symmetry properties of the flag.

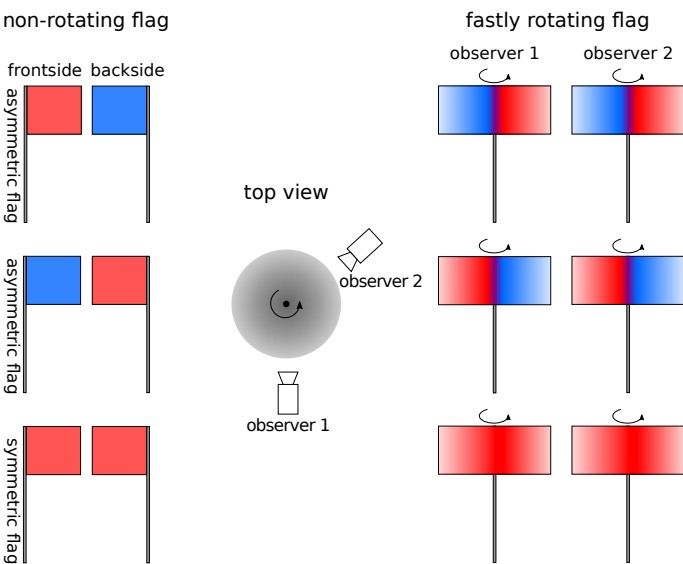

**Figure 3.** Schematic showing that rotation results in a rotational symmetry around the flagpole (axis). The actual image that we see depends on the symmetry properties of the flag (object).

## 3   Basic setup and shape data

The scattering calculations are computationally demanding in view of the computation time and the amount of data. Therefore, we have to compromise in terms of the accuracy of the resulting scattering data. Considering the measurement errors of existing and upcoming passive MW and SubMM sensors, which are in the order of $\mathcal{O}(1\,\mathrm{K})$, and the brightness temperature depression due to scattering of frozen hydrometeors, which is typically $< 100\,\mathrm{K}$, we aim for an accuracy of the scattering database in general in the order of a few percent with respect to the scattering properties of the assumed scatterer shapes. This aim is the

guideline for our scattering calculation.

For the scattering calculations ADDA version 1.2 was used. ADDA is a DDA implementation of Yurkin and Hoekstra (2011). The basic idea of DDA is to represent the particle by a discrete set of electric dipoles. To calculate the scattering, ADDA iteratively solves the linear system

$$\boldsymbol{\alpha}_i \boldsymbol{P}_i - \sum_{i \neq j} \boldsymbol{H}_{ij} \boldsymbol{P}_j = \boldsymbol{E}_{inc,i} \qquad (10)$$

with $i, j$ the dipole indices, $\boldsymbol{\alpha}_i$ the dipole polarizability, $\boldsymbol{P}_i$ the unknown dipole polarization, $\boldsymbol{H}_{ij}$ the interaction term and $\boldsymbol{E}_{inc,i}$ the incident electric field. The resulting scattering quantities of ADDA are derived from the solution of the dipole polarization $\boldsymbol{P}_l$, for details see Yurkin and Hoekstra (2011). The iteration is stopped when the relative norm of the residuals $\epsilon$ is less than a user specified value. The relative norm of the residuals $\epsilon$ is essentially the relative difference between the left-hand side and the right-hand side of Eq. 10. To reduce the computation time in view of our desired accuracy for the scattering

database, we set the relative norm of the residuals to

$$\epsilon = 10^{-2}. \tag{11}$$

For further details of the DDA method, see Yurkin and Hoekstra (2011) and the references therein.

ADDA can simulate the scattering of totally randomly oriented particles and the scattering of particles with a fixed but arbitrary orientation. The internal averaging method of ADDA is not suitable for our approach to simulate azimuthally oriented
particles. Instead, we developed an averaging approach that involves integration over a set of DDA calculations at different angles, which is explained in Sect. 4.

For DDA simulations it is important that the size of the dipoles is small compared to the wavelength and to any structural length within the scatterer (Yurkin and Hoekstra, 2011). For all particles considered in our study holds

$$|m| kd < \frac{1}{2} \tag{12}$$

with $m$ the refractive index of ice, $k$ the angular wavenumber and $d$ the dipole size. With the microwave refractive index of ice this results in $\approx 22$ dipoles per wavelength. Furthermore, all simulated particles consist of at least $1,000$ dipoles so that small particles are reasonably resolved.

Following Eriksson et al. (2018), we organize the different particle shapes as habits. A habit is defined as a set of particles of different sizes with a common basic morphology, roughly following a mass-size relationship. In this work we consider two
different types of frozen hydrometeor habits:

- plate type 1, which is a solid hexagonal plate-like single crystal, and

- large plate aggregate, which consists of several solid hexagonal plates aggregated to one particle.

Fig. 4 shows some different sized particles of both habits as example. The shape data including the actual dipole grids for ADDA were taken from the database of Eriksson et al. (2018). The mass-size relationship is defined as

$$m = a \left( \frac{D}{D_0} \right)^b \tag{13}$$

with $m$ the particle mass, $D$ the maximum diameter, the unit diameter $D_0 = 1\,\mathrm{m}$ and the parameters $a$, $b$. The maximum diameter is defined as the diameter of the minimum circumscribed sphere of a particle. Table 1 shows for each habit the size range and the values of the parameters $a$ and $b$. For the plate type 1 habit, 51 differently sized particles were simulated. The size range is between $10\,\mu\mathrm{m}$ and $2,596\,\mu\mathrm{m}$ volume equivalent diameter, which corresponds to maximum diameters between
$13\,\mu\mathrm{m}$ and $10,000\,\mu\mathrm{m}$. The volume equivalent diameter is defined as the diameter of a solid ice sphere with the same mass as the particle. For the large plate aggregate habit, 18 differently sized particles were simulated. The size range is between $197\,\mu\mathrm{m}$ and $4,563\,\mu\mathrm{m}$ volume equivalent diameter, which corresponds to maximum diameters between $349\,\mu\mathrm{m}$ and $22,860\,\mu\mathrm{m}$. The plate type 1 habit and the large plate aggregate habit in our study have slightly different sizes than the corresponding habits

**Table 1.** Overview of the selected habits. $a$ and $b$ are the parameters of the mass-size relationship (Eq. 13), $D_{veq}$ is the volume equivalent diameter and $D_{max}$ is the maximum diameter. ID is the identification number from the database of Eriksson et al. (2018).

| habit name | ID | type | $a$ [kg] | $b$ | No. of sizes | $D_{veq}$ [µm] | $D_{max}$ [µm] |
|---|---|---|---|---|---|---|---|
| plate type 1 | 9 | single crystal | 0.76 | 2.48 | 51 | $10 - 2,596$ | $13 - 10,000$ |
| large plate aggregate | 20 | aggregate | 0.21 | 2.26 | 18 | $197 - 4,563$ | $349 - 22,860$ |

**Table 2.** The frequencies of the scattering calculations. Except for 35.6 GHz, the channels $\geq 18.6$ GHz are organized in channel sets, see text.

| Channel set | 1 | 2 | 3 | 4 | 5 | 6 | 7 | 8 | 9 | 10 | 11 | 12 |
|---|---|---|---|---|---|---|---|---|---|---|---|---|
| Freq. | 18.6 | 31.3 | 50.1 | 88.8 | 115.3 | 164.1 | 175.3 | 228 | 314.2 | 439.3 | 657.3 | 862.4 |
| [GHz] | 24 | 31.5 | 57.6 | 94.1 | 122.2 | 166.9 | 191.3 | 247.2 | 336.1 | 456.7 | 670.7 | 886.4 |

Other frequencies [GHz]:

1, 1.4, 3, 5, 7, 9, 10, 10.65, 13.4, 15, 35.6

in Eriksson et al. (2018), because an older version of shape data than in Eriksson et al. (2018) was used and given the high
computational costs of the scattering calculation a recalculation was not feasible. Therefore, we included the shape files in the database. For details on the particle shape data the reader is referred to Eriksson et al. (2018).

In this work we follow the approach of Eriksson et al. (2018) for the temperature and frequency selection. The selected frequency range of the scattering calculation consists of 35 frequencies between 1 GHz and 864 GHz. Most selected frequencies are organized to include channel sets of existing and planned submillimeter and microwave radiometers to provide frequency
coverage at the part of the spectrum, where today and in the future observations are done and will be done. Table 2 shows the selected frequencies. It is important to note that outside of the defined channel ranges, the database must be used with care. Interpolation between the channels can be done, but it must judged from case to case. Interpolating at 170 GHz is less likely to be an issue, as the separation between channels 6 and 7 is small, but interpolating at 550 GHz is likely to be an issue due to the large separation between channels 10 and 11. Due to a rounding mistake when the simulation was set up, the frequencies
of the plate type 1 habit slightly deviate from the frequencies of the large plate aggregate habit by at maximum 0.5 GHz. The selected temperatures are 190 K, 230 K, and 270 K. Following Eriksson et al. (2018), the refractive index of ice is calculated using the model of Mätzler (2006).

## 4   Scattering calculations

In general, the scattering matrix $\boldsymbol{Z}$ of a non-spherical particle depends on the incidence direction $(\theta_{inc}, \phi_{inc})$, the scattering
direction $(\theta_s, \phi_s)$ and the particle orientation described by the three Euler angles $\alpha$, $\beta$ and $\gamma$. The same holds for the extinction matrix $\boldsymbol{K}$ except that it is independent of the scattering directions. The rotation of a particle is equivalent to the inverse rotation of the incidence direction. This means that it is equivalent, if the scattering of a particle is calculated for any incidence angle at a

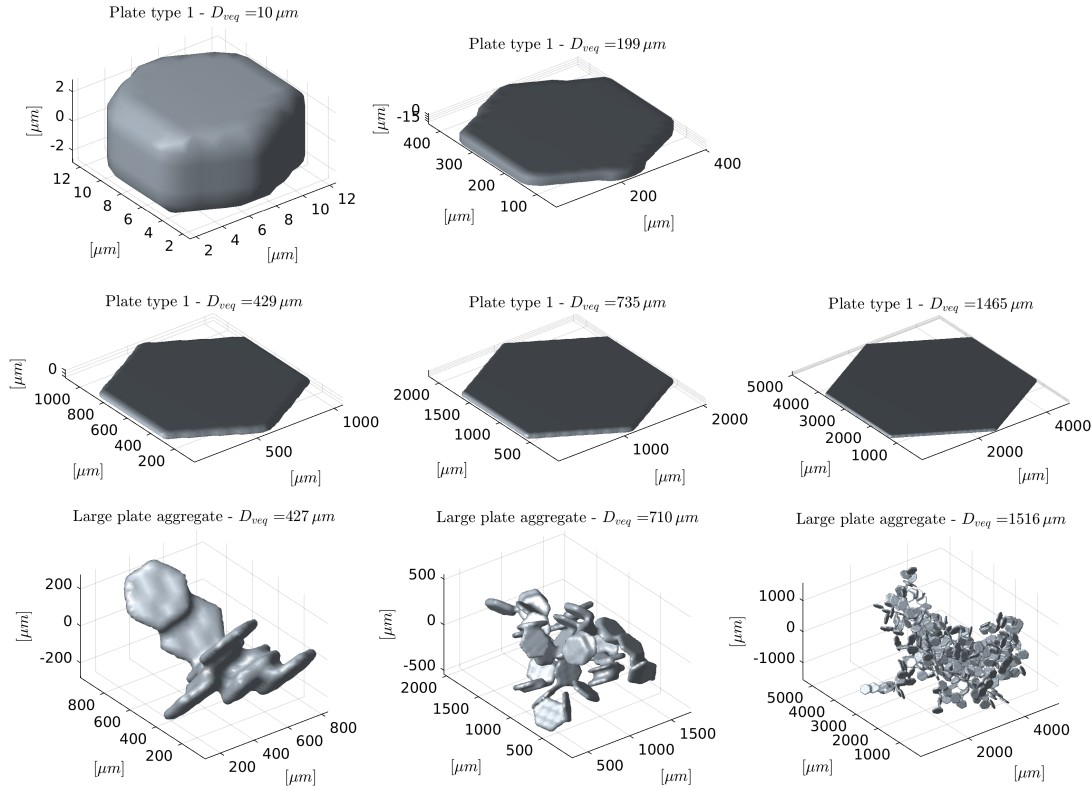

**Figure 4.** Example scatterer shapes.

fixed orientation, or if the scattering of a particle is calculated for any orientation but at a fixed incidence angle. This equivalence is the key point in our approach. The scattering is therefore calculated for any incidence direction and scattering direction and

the particle orientation is kept fixed. The orientation averaging is calculated by rotating the incidence and scattering direction according to the particle orientation. With ADDA it is only possible to calculate the scattering properties for a finite set of incidence and scattering directions. Hence, the scattering matrix and the extinction matrix are calculated for a set of different incidence directions and scattering directions (only scattering matrix). The result is the scattering matrix and the extinction matrix for finite set of incidence and scattering directions, which are fixed to the particle, see Fig. 5 a. For a specific orientation

of the particle, the set of incidence and scattering directions are rotated according to the orientation of the particle, see Fig. 5 b.

The actual results of an ADDA calculation are the scattering amplitude matrix and the Mueller matrix for a desired incidence direction and a grid of scattering directions, whereas we are interested in extinction matrix and scattering matrix. The relationship between the scattering amplitude matrix and the extinction matrix and between the Mueller matrix and the scattering matrix are explained in the following paragraphs. Difficulties arise from the fact that the matrices are defined in different

coordinate systems. In the database, the scattering matrix and the extinction matrix are defined in the laboratory system. The

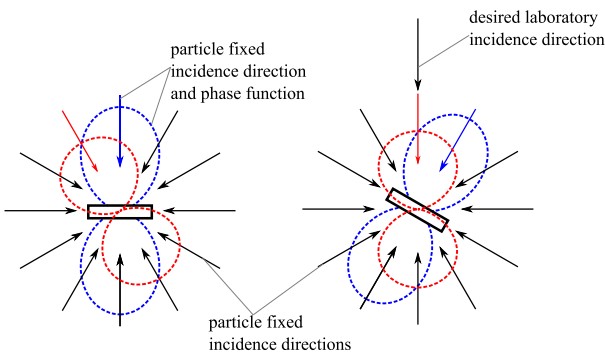

a) scan the non rotated particle    b) rotate particle to desired orientation

desired laboratory
incidence direction

particle fixed
incidence direction
and phase function

particle fixed
incidence directions

**Figure 5.** Schematic drawing of the calculation of the single scattering properties. (left) the non rotated particle with the incidence and scattering directions fixed to the particle. (right) the rotated particle and the rotated incidence and scattering directions.

extinction matrix that results from the scattering amplitude matrix and the Mueller matrix are defined in the coordinate system that is defined by the incidence direction and the particle system, from here on called wave reference system. Due to the relation to the particle system, the wave reference system rotates if the particle (particle system) rotates. Therefore, the main part of our averaging approach consists essentially of transformations from one coordinate system to another coordinate system.

The extinction matrix $K$ depends on the scattering amplitude matrix for the forward direction ($\theta_{inc} = \theta_s$, $\phi_{inc} = \phi_s$, Mishchenko et al. 2002)

$$K = \frac{2\pi}{k} \begin{pmatrix} \mathrm{Im}\,(S_{11} + S_{22}) & \mathrm{Im}\,(S_{11} - S_{22}) & -\mathrm{Im}\,(S_{12} + S_{21}) & \mathrm{Re}\,(S_{21} - S_{12}) \\ \mathrm{Im}\,(S_{11} - S_{22}) & \mathrm{Im}\,(S_{11} + S_{22}) & \mathrm{Im}\,(S_{21} - S_{12}) & -\mathrm{Re}\,(S_{12} + S_{21}) \\ -\mathrm{Im}\,(S_{12} + S_{21}) & -\mathrm{Im}\,(S_{21} - S_{12}) & \mathrm{Im}\,(S_{11} + S_{22}) & \mathrm{Re}\,(S_{22} - S_{11}) \\ \mathrm{Re}\,(S_{21} - S_{12}) & \mathrm{Re}\,(S_{12} + S_{21}) & -\mathrm{Re}\,(S_{22} - S_{11}) & \mathrm{Im}\,(S_{11} + S_{22}) \end{pmatrix} \tag{14}$$

with the scattering amplitude matrix

$$S = \begin{pmatrix} S_{11} & S_{12} \\ S_{21} & S_{22} \end{pmatrix} = \frac{1}{-ik} \begin{pmatrix} s_2 & s_3 \\ s_4 & s_1 \end{pmatrix}, \tag{15}$$

$k$ the angular wavenumber and $s_j$ scattering amplitude matrix element of ADDA. The scattering amplitude matrix is a complex matrix and operates on the complex electric field, whereas the extinction, the scattering, and the Mueller matrix operate on the Stokes vector, which is a real-valued vector. Between the scattering matrix $Z$ and the Mueller matrix $M$, which are both real $4 \times 4$ matrices, the following linear relationship holds

$$Z = \frac{1}{k^2} L_s M L_i \tag{16}$$

with the Stokes rotation matrices $L_i$ and $L_s$ (Mishchenko et al., 2002). The Stokes rotation matrices transform the Mueller matrix from the wave reference system to the laboratory system. The stokes rotation matrices $L_{i,s}$ are defined in Sect. D. Due

to the linear relationship, it does not matter whether first the Mueller matrix is transformed to a scattering matrix and then the scattering matrix is averaged or vice versa. Instead of transforming every calculated Mueller matrix into the scattering matrix, the averaging will be done for the Mueller matrix and lastly the averaged Mueller matrix is transformed to the scattering matrix, which is described in Appendix D.

Each Mueller matrix element $M_{ij}(\theta_{inc}, \phi_{inc}, \theta'_s, \phi'_s)$, which has a scattering direction grid spacing of $1°$, is expanded as a spherical harmonics series over the scattering angles $\theta'_s$ and $\phi'_s$ (see Appendix E) to efficiently store the results of the ADDA calculation. The prime denotes that the angles are related to the wave reference system and not to the laboratory system. To reduce the amount of data, the spherical harmonic series are truncated to the number of coefficients for which the mean square error between the series expansion and the original representation is less than $0.5\%$ of the standard deviation of the $M_{11}$ element over the scattered direction. Relating the truncation to the $M_{11}$ element results on average that after the truncation features of the other Mueller matrix elements are still resolved if their magnitude is greater than the truncation error of $M_{11}$. This allows to resolve the relevant features given the desired accuracy of the scattering database and reduces the amount of data by up to two orders of magnitude.

For each incidence direction, ADDA automatically calculates the Mueller matrix for a desired regular grid of polar angles and azimuth angles. A regular grid of polar and azimuth angles has the property that the grid spacing at the pole is much finer than at the equator. For the incidence angles, a regular grid of polar angles and azimuth angles are disadvantageous, because for the incidence angle an isotropic sampling is needed, but the distribution of the directions of a regular grid of polar angles and azimuth angles is not isotropic. Therefore, an icosahedral grid is used, which is shown in Fig. 6. An icosahedral grid is almost isotropic. An icosahedral grid consists of equilateral triangles, which have the same size, and the distances between two neighboring vertices (grid points) is everywhere the same. This makes the icosahedral grid convenient for grid refinement and adjusting the grid size for the needed accuracy. An icosahedral grid can be set up by recursively bisecting the edges of an icosahedron and projecting the new vertices on a sphere. Such an icosahedral grid consists of

$$N_v = 10 \cdot (2l)^2 + 2 \tag{17}$$

vertices and

$$N_t = 20 \cdot (2l)^2 \tag{18}$$

triangles with $l$ the refinement level. The vertex coordinates of the icosahedral grid are the set of incidence directions. For more details on icosahedral grids, see for example Satoh (2014).

The orientation averaged Mueller matrix $\boldsymbol{M}_{aro}$ is

$$\boldsymbol{M}_{aro}(\theta_{inc}, \theta'_s, \phi'_s, \beta) = \int_0^{2\pi} \int_0^{2\pi} p_\alpha(\alpha) p_\gamma(\gamma) R^*_{\alpha\beta\gamma}(\boldsymbol{M}) \, \mathrm{d}\alpha \, \mathrm{d}\gamma \tag{19}$$

and orientation averaged extinction matrix $\boldsymbol{K}_{aro}$ is

$$\boldsymbol{K}_{aro}(\theta_{inc}, \beta) = \int_0^{2\pi} \int_0^{2\pi} p_\alpha(\alpha) p_\gamma(\gamma) R^*_{\alpha\beta\gamma}(\boldsymbol{K}) \, \mathrm{d}\alpha \, \mathrm{d}\gamma \tag{20}$$

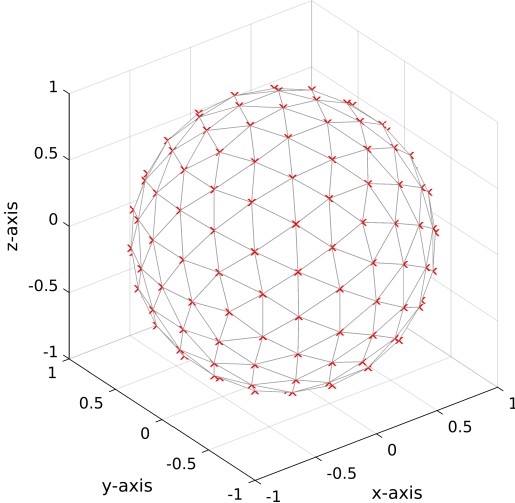

**Figure 6.** Example of an icosphere grid with 162 vertices. Each gridpoint represent an incoming angle for which a DDA calculation is performed. This type of configuration ensures that the grid density is isotropic, making the overall calculations more efficient (a standard polar grid would be inefficient, since it yields an excessive amount of angles around the 'North and South poles").

The rotation operator $R^*_{\alpha\beta\gamma}$ rotates the Mueller and the extinction matrix according to the desired orientation, which is explained in Appendix. B. The needed interpolation is done by using a barycentric interpolation for triangles, which is explained in Appendix C. Afterwards, the averaged Mueller matrix $\boldsymbol{M}_{aro}(\theta_{inc}, \theta'_s, \phi'_s, \beta)$ is transformed into the scattering matrix $\boldsymbol{Z}_{aro}$ using Eq. 16, which is explained in Appendix. D. As mentioned in Sect. 2.2, the resulting scattering matrix $\boldsymbol{Z}_{aro}$ is in general not symmetric, as this depends on the actual particle. The scattering matrix $\boldsymbol{Z}_{aro}$ is symmetric if it is averaged with its own mirrored version in which it is reflected relative to the plane of incidence direction and laboratory Z-axis. This is equivalent to having simulated the scattering of the desired particle and its mirrored version, in which it is reflected by a plane that includes the laboratory Z-axis, see Mishchenko et al. (2002) or van de Hulst (1981) for further details on the symmetry of the scattering matrix.

The actual scattering calculations are done iteratively. For each particle, the scattering calculation begins with 12 incidence angles (refinement level $l = 0$). With each additional refinement level $l$ the number of incidence angles increases according to Eq. 17 roughly by a factor of four. With each iteration step the edges of the triangles of the icosahedral grid are bisected creating new vertices (incidence angles). This means that the incidence angles of the previous iteration are part of the grid for the current iteration. As such, only about $\frac{3}{4}$ of the number of incidence angles have to be calculated for each iteration step. The iteration stops when

$$\frac{\delta_{l,l-1}}{\delta_{l-1,l-2}} \leq 10^{-2}. \tag{21}$$

The change $\delta_{l,l-1}$ between the current iteration step $l$ and the previous iteration step is defined as the summed root mean square differences between the upper left block of the orientation averaged extinction matrix of iteration step $l$ and $l-1$ for five

different tilt angles $\beta$ and ten incidence angles $\theta_{inc}$. Depending on the particle size and shape, between 162 and 2562 incidence angles were used.

To test our approach, the scattering of azimuthally randomly oriented prolate ellipsoids with an aspect ratio of 0.5 for several size parameters were simulated and compared with results from T-matrix calculations. The overall differences in view of the extinction matrix and the scattering matrix were in the order of a few percent. Strictly speaking, this test shows only the differences to T-matrix simulation of azimuthally randomly oriented prolate ellipsoids with an aspect ratio of 0.5. Nonetheless, it gives an idea of the overall accuracy of the scattering simulations. Therefore, we expect that the overall accuracy of the scattering simulations is about the same of magnitude.

The methodology to calculate the scattering matrix and the extinction matrix can be summarized as:

1. DDA calculations: A set of DDA runs is performed over an icosahedral angle grid of incidence directions, demonstrated in Fig. 6. This type of grid ensures that the angle density is isotropic and increases the efficiency.

2. Represent the Mueller matrix elements of each ADDA run in a spherical harmonics series and truncate them to reduce the amount of data.

3. Averaging: Azimuthally averaged Mueller matrices $\boldsymbol{M}_{aro}(\theta_{inc}, \theta'_s, \phi'_s, \beta)$ and extinction matrices $\boldsymbol{K}_{aro}(\theta_{inc}, \beta)$ for a set of tilt angles $\beta$ and polar incidence angles $\theta_{inc}$ are calculated by integrating the Mueller and extinction matrices over the Euler angles $\alpha$ and $\gamma$.

4. Transformation: The averaged Mueller matrices are transformed to averaged scattering matrices $\boldsymbol{Z}_{aro}$.

## 5 Results of the scattering simulations

In this section we give an overview of the scattering simulations and show some example results. 51 sizes of plate type 1 (hexagonal plate) and 18 sizes of large plate aggregates for 35 frequencies and 3 temperatures were simulated. The simulations were conducted on DKRZ's (Deutsches Klimarechenzentrum) supercomputer Mistral. This took about $1.6 \cdot 10^6$ core hours on Intel Xeon E5-2695V4 processors with a clock rate of $2.1\,\mathrm{GHz}$. The amount of data of the scattering calculations is huge. Whereas the scattering matrix $\boldsymbol{Z}_{\mathrm{tro}}(\Theta)$ for total random orientation depends on one angle, the scattering matrix $\boldsymbol{Z}_{\mathrm{aro}}(\theta_{inc}, \theta_s, \phi_s)$ for azimuthal random orientation depends on three angles. Furthermore, the tilt angle $\beta$ adds an additional dimension. This leads to an up to three orders of magnitude larger amount of data. To reduce the computational time, the residual relative norm, which is the stopping criterion of ADDA's iterative solver, was set to $10^{-2}$ following Eriksson et al. (2018). The Mueller and the scattering matrices for a given incidence angle were represented in a truncated spherical harmonics series to reduce the amount of data. Even then, the total size of the data from the DDA simulations is about $1.5\,\mathrm{TB}$. Due to the orientation averaging the amount of data reduces to about $0.18\,\mathrm{TB}$.

The orientation averaging is done for a finite set of incidence and tilt angles. The incidence angles $\theta_{inc}$ span a range from $0°$ to $180°$ with a $5°$ spacing and the tilt angles $\beta$ span a range from $0°$ to $90°$ for plate type 1 and from $0°$ to $180°$ for large

plate aggregates with a $10°$ spacing. The tilt angle range for plate type 1 is confined to $90°$, because of its mirror symmetry to the x-y plane. In this case, it holds for the scattering matrix $\boldsymbol{Z}_{\mathrm{aro}}$ and the extinction matrix $\boldsymbol{K}_{\mathrm{aro}}$ that

$$\begin{aligned}
\boldsymbol{Z}_{\mathrm{aro}}\left(\theta_{inc},\theta_s,\phi_s,\beta\right) &= \boldsymbol{Z}_{\mathrm{aro}}\left(\theta_{inc},\theta_s,\phi_s,\pi-\beta\right) \\
\boldsymbol{K}_{\mathrm{aro}}\left(\theta_{inc},\beta\right) &= \boldsymbol{K}_{\mathrm{aro}}\left(\theta_{inc},\pi-\beta\right).
\end{aligned} \tag{22}$$

The scattering database with the orientation averaged data is publicly available from Zenodo (https://doi.org/10.5281/zenodo.3463003). The data from the DDA simulations in truncated spherical harmonics representation is available upon request from us. The scattering database is organized so that the Python 3 interface of the database of Eriksson et al. (2018) can be used to extract and interact with the data. The scattering database additionally includes the absorption vector $\boldsymbol{a}$ for each incidence and tilt angle. The $i$-th component of the absorption vector is

$a_i\left(\theta_{inc},\beta\right) = K_{aro,i1}\left(\theta_{inc},\beta\right) - \int\limits_{0}^{2\pi}\int\limits_{0}^{\pi} Z_{aro,i1}\left(\theta_{inc},\theta_s,\phi_s,\beta\right)\mathrm{d}\phi_s\mathrm{d}\theta_s$           (23)

with $K_{aro,i1}$ and $Z_{aro,i1}$ being the $i$-th component of the first column of the extinction matrix $\boldsymbol{K}_{\mathrm{aro}}$ and scattering matrix $\boldsymbol{Z}_{\mathrm{aro}}$ (Mishchenko et al., 2000), respectively.

     In the following analysis we will not address the absorption vector, because it is derived directly from the extinction and scattering matrix and is just added to the database for convenience.

**5.1    Extinction matrix and asymmetry parameter**

The orientation averaging (Eq. 20) reduces Eq. 14 to

$$\boldsymbol{K}_{aro} = \frac{2\pi}{k}\begin{pmatrix}
\mathrm{Im}\left(S_{11}+S_{22}\right) & \mathrm{Im}\left(S_{11}-S_{22}\right) & 0 & 0 \\
\mathrm{Im}\left(S_{11}-S_{22}\right) & \mathrm{Im}\left(S_{11}+S_{22}\right) & 0 & 0 \\
0 & 0 & \mathrm{Im}\left(S_{11}+S_{22}\right) & \mathrm{Re}\left(S_{22}-S_{11}\right) \\
0 & 0 & -\mathrm{Re}\left(S_{22}-S_{11}\right) & \mathrm{Im}\left(S_{11}+S_{22}\right)
\end{pmatrix} \tag{24}$$

with $S_{ii}$ the scattering amplitude matrix elements (Eq. 15) and $k$ the angular wavenumber. Whereas the extinction matrix has seven independent entries in general, the extinction matrix for azimuthal random orientation has only three independent entries

that depend on the incidence angle $\theta_{inc}$ and the tilt angle $\beta$. For total random orientation the extinction matrix has only one independent entry.

     Fig. 7 and Fig. 8 show the three independent entries of the extinction matrix ($K_{11}$, $K_{21}$, and $K_{43}$) of plate type 1 and large plate aggregate at $671\,\mathrm{GHz}$ for several tilt angles $\beta$ and size parameters $x$

$$x = ka_{eq} = \frac{2\pi a_{eq}}{\lambda} = \frac{\pi D_{eq}}{\lambda} \tag{25}$$

with $a_{eq}$ the volume equivalent frozen radius, $D_{eq}$ the volume equivalent frozen diameter and $\lambda$ the wavelength. For the large plate aggregate habit only size parameters $x > 3$ are shown, because for smaller sizes it is practically the same as plate type

1. The extinction matrix elements in Fig. 7 and Fig. 8 are normalized by the extinction cross section $K_{tro}$ for total random orientation of the specific shape. Using Eq. 5 the extinction cross section for total random orientation $K_{tro}$ is

$$K_{tro} = \int_0^\pi p_\beta(\beta) K_{aro,11}(\theta_{inc}, \beta) \, \mathrm{d}\beta. \tag{26}$$

For the large plate aggregate we skip the tilt angles $\beta > 90°$ in Fig. 8, because for $\beta > 90°$ the results are the same as for $\beta < 90°$ but mirrored around $\theta_{inc} = 90°$. Due to the mirror symmetry to the x-y plane of the hexagonal plates, the curves shown in Fig. 7 are symmetric relative to $\theta_{inc} = 90°$.

For the plate type 1 habit the effect of orientation and incidence angle results in differences of up to $50\%$ of the $K_{aro,11}$ element compared to total random random orientation, whereas for the large plate aggregate habit the biggest differences

are at maximum about $15\%$. The biggest differences occur for tilt angles of $0°$ and $90°$ when looking from the top/bottom ($\theta_{inc} = 0°, 180°$) and from the side ($\theta_{inc} = 90°$). Depending on the size parameter, shape and magnitude of the curve change. For example, the maximum for the plate type 1 habit occurs at tilt angle $\beta = 0°$ and incidence angles of $0°$ and $180°$ for $x \lesssim 1$ and $x \approx 10$, whereas it occurs at an incidence angle of $90°$ for $x \approx 3$ and $x \approx 5$. The large plate aggregate habit shows a similar behavior albeit with much lower magnitude.

The $K_{aro,21}$ matrix element describes the extinction of the polarization difference between vertical and horizontal polarization and the $K_{aro,43}$ matrix element the extinction of polarization difference between the $+45°$ and $-45°$ polarization. For total random orientation, these matrix elements are zero, which is indicated by the gray line in Fig. 7 and Fig. 8. For the plate type 1 habit the $K_{aro,21}$ and the $K_{aro,43}$ matrix element show a strong dependency on the tilt angle and the incidence angle, which reduces with increasing size parameter. Except when looking from the top/bottom ($\theta_{inc} = 0°, 180°$) both elements are

non-zero. For the large plate aggregate habit the $K_{aro,21}$ and the $K_{aro,43}$ matrix element are practically zero, showing only small deviations from zero for $x \gtrsim 3$.

For $x \approx 1.4$ and tilt angle $\beta = 0°$ the results for the plate type 1 agree qualitatively with the results of Adams and Bettenhausen (2012) for azimuthally randomly oriented hexagonal plates with tilt angle $\beta = 0°$ and a similar size parameter but at a different frequency.

The asymmetry parameter describes the distribution between forward scattering and backscattering and gives an overview of the scattering behavior. For example, $g = 0$ means forward scattering and backscattering are of equal strength, whereas $g = 1$ and $g = -1$ mean only forward scattering and only backscattering, respectively. The asymmetry parameter for azimuthal random orientation is

$$g_{aro}(\theta_{inc}, \beta) = \frac{1}{2} \int_0^{2\pi} \int_0^\pi \cos(\theta_s - \theta_{inc}) Z_{\mathrm{aro},11}(\theta_{inc}, \theta_s, 0, \phi_s, \beta) \, \mathrm{d}\phi_s \mathrm{d}\theta_s \tag{27}$$

with $Z_{aro,11}$ being the $(1,1)$-element of the scattering matrix $\boldsymbol{Z}_{aro}$. The asymmetry parameter is shown in Fig. 7 and Fig. 8. For the different tilt angles the asymmetry parameters are centered around the asymmetry parameter $g_{tro}$ for total random orientation, which is shown as a gray line. The asymmetry parameter $g_{tro}$ is calculated by integrating $g_{aro}(\theta_{inc}, \beta)$ over the tilt

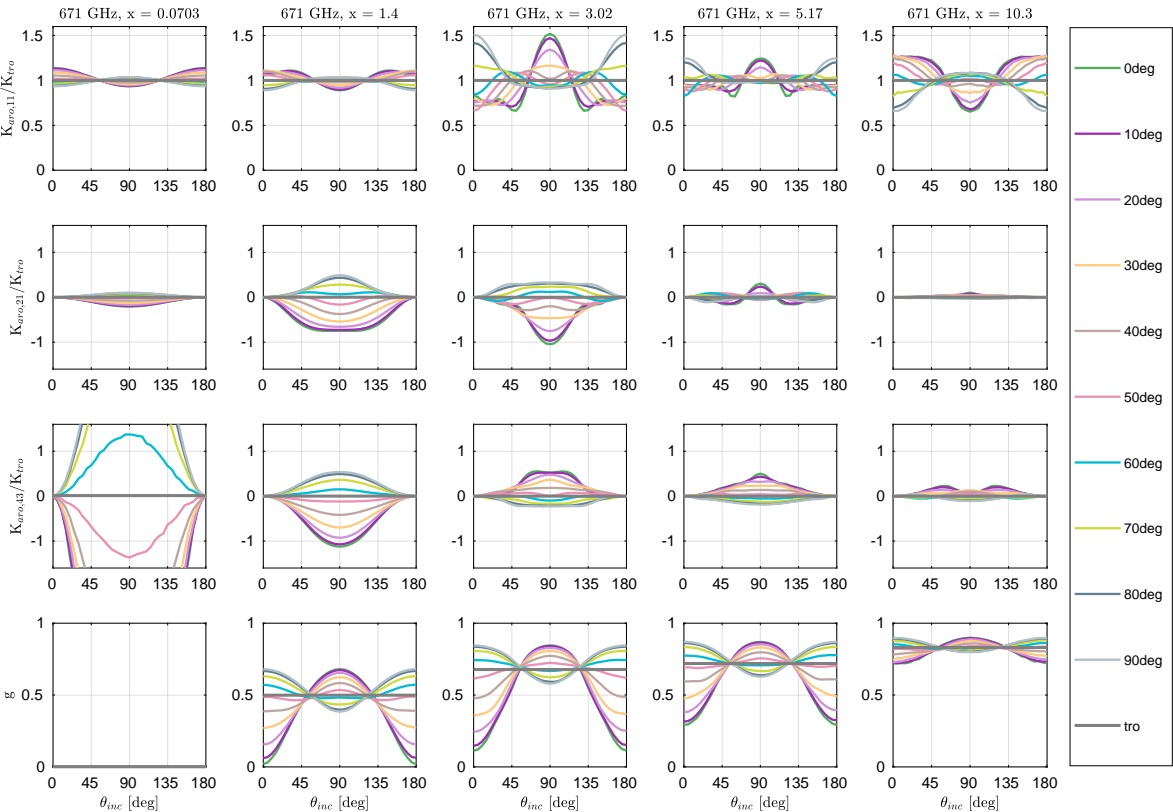

**Figure 7.** Extinction matrix elements $K_{aro,ij}$ normalized by the extinction cross section for total random orientation and the asymmetry parameter $g$ of plate type 1 (hexagonal plate) for different size parameter $x$ at 671 GHz as function of incidence angle $\theta_{inc}$ for several tilt angles $\beta$. The gray lines denote total random orientation. The shapes of the scatterers are shown in Fig. 4.

angle $\beta$ similar to Eq. 26. For $x \ll 1$, the asymmetry parameter $g_{tro}$ is zero indicating symmetric forward and backward scattering as expected for Rayleigh scattering. With increasing size parameter forward scattering increases. The azimuthal random orientation asymmetry parameter $g_{aro}$ for the large plate aggregate habit deviates slightly from the asymmetry parameter $g_{tro}$ with changing tilt angle $\beta$, whereas for the plate type 1 habit it deviates strongly from the asymmetry parameter $g_{tro}$, especially for $1 < x < 6$. For example, at $\beta = 0°$ and incidence angles of $0°$ and $180°$ for $x = 1.4$ the scattering in forward and backward direction is almost symmetric but at $\beta = 90°$ the scattering in forward direction is much stronger than in backward direction.

## 5.2 Scattering matrix

The scattering matrix of a particle describes the angular distribution of the scattered radiation in relation to the incidence direction of the incoming radiation. For unpolarized incoming radiation, the $Z_{j1}$-element with $j = \{1, ..., 4\}$ describes the

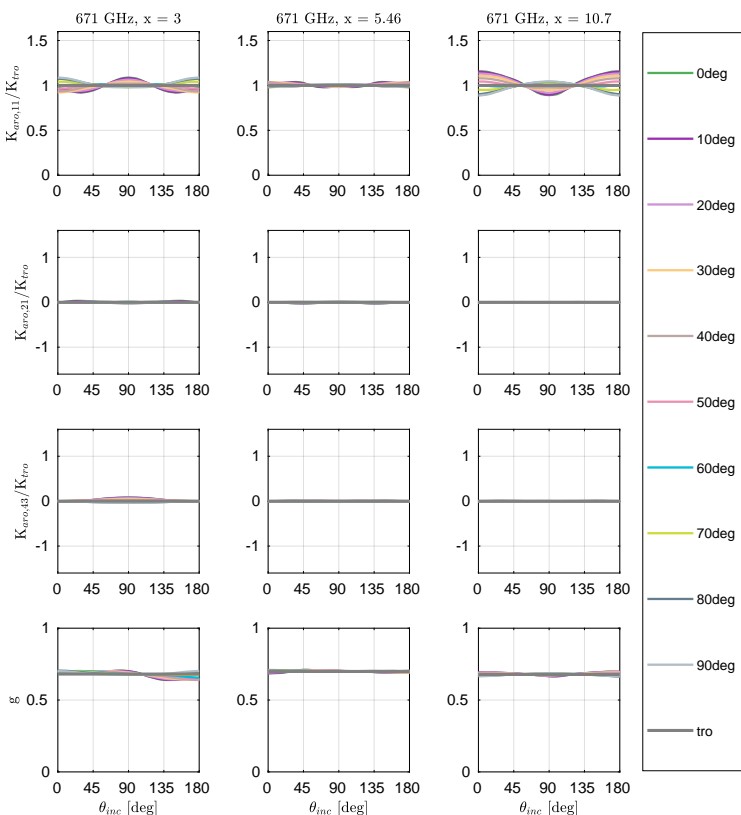

**Figure 8.** Extinction matrix elements $K_{aro,ij}$ normalized by the extinction cross section for total random orientation and the asymmetry parameter $g$ of large plate aggregate (hexagonal plate aggregate) for different size parameter $x$ at $671\,\mathrm{GHz}$ as function of incidence angle $\theta_{inc}$ for several tilt angles $\beta$. The gray lines denote total random orientation. The shapes of the scatterers are shown in Fig. 4.

angular distribution of the scattered radiation field. For example, the $Z_{11}$-element gives the angular distribution of the scattered intensity (I component of the Stokes vector), whereas the $Z_{21}$-element determines how and where the scattered radiation is horizontally and vertically polarized (Q component of the Stokes vector). Negative $Z_{21}$ values mean that the horizontal polarization dominates and vice versa. For polarized radiation, the $j$-th component of the scattered radiation field depends additionally on the coupling with the other components of the incoming Stokes vector, which is described by the $Z_{ji}$-element with $i = \{2, 3, 4\}$.

After the orientation averaging, the resulting scattering properties possess a rotational symmetry relative to the laboratory z-axis. The scattering matrix $\boldsymbol{Z}_{aro}$ (Eq. 19, D1) depends for tilt angle $\beta$ on the polar incidence angle $\theta_{inc}$, the polar scattering angle $\theta_s$ and the scattering azimuth angle $\phi_s$. In contrast to the scattering matrix of totally randomly oriented particles that depends only on the scattering angle $\Theta$. The different tilt angles $\beta$ result in different effective shapes and therefore different

scattering matrices. The impact of the tilt angle $\beta$ depends also on the incidence direction and is different for the different scattering matrix elements.

As an example, Fig. 9 shows the upper left block of the normalized scattering matrix $\hat{\boldsymbol{Z}}_{\text{aro}}(\theta_{inc}, \theta_s, \phi_s)$ of plate type 1 for size parameter $x \approx 3$ at $671\,\text{GHz}$ and for several incidence angles $\theta_{inc}$ and tilt angles $\beta$. The normalized scattering matrix $\hat{\boldsymbol{Z}}_{\text{aro}}(\theta_{inc}, \theta_s, \phi_s)$ is

$$\hat{\boldsymbol{Z}}_{\text{aro}}(\theta_{inc}, \theta_s, \phi_s) = 4\pi \frac{\boldsymbol{Z}_{aro}}{\int_0^{2\pi} \int_0^{\pi} \boldsymbol{Z}_{\text{aro}}(\theta_{inc}, \theta_s, \phi_s)\, \mathrm{d}\phi_s \mathrm{d}\theta_s}\ . \tag{28}$$

We show only the upper left block, because these are the most relevant entries of the scattering matrix considering the present spaceborne microwave and submillimeter wave sensors. However, all 16 elements are calculated. At incidence direction $\theta_{inc} = 0°$, the $\hat{Z}_{11}$- and $\hat{Z}_{22}$-element differ strongly between the different tilt angles $\beta$. Especially in the backscattering direction, they strongly decrease with increasing tilt angle $\beta$. The $\hat{Z}_{21}$- and $\hat{Z}_{12}$-element show only slight differences between the different tilt angles. The $\hat{Z}_{11}$-element decreases at backscattering direction with increasing tilt angle, but it is fairly constant at the forward direction. This results, in total, in an increased forward direction, which is also shown by the asymmetry parameter $g_{aro}$ in Fig. 7. Within the Rayleigh regime ($x \ll 1$, not shown), the influence of the tilt angle $\beta$ on the normalized scattering matrix $\hat{\boldsymbol{Z}}_{aro}$ is negligible at incidence direction $\theta_{inc} = 0°$.

For non nadir/zenith incidence directions the $\hat{Z}_{21}$- and $\hat{Z}_{12}$- element, as well the other scattering matrix elements, differ strongly for different tilt angle $\beta$. For example, the $\hat{Z}_{21}$- and $\hat{Z}_{12}$- elements have negative peaks at $\theta_s = 180° - \theta_{inc}$ and $\phi_s = 0°$ for tilt angle $\beta = 0°$, which means that unpolarized radiation scattered in this direction is horizontally polarized. There is no peak at this scattering direction for tilt angle $\beta = 50°$ or $\beta = 90°$. For tilt angle $\beta = 50°$ there is a negative peak at $\theta_s = \theta_{inc}$ and for tilt angle $\beta = 90°$ there is a positive peak at $\theta_s = \theta_{inc}$. The negative peaks of the $\hat{Z}_{21}$- and $\hat{Z}_{12}$- element at $\theta_s = 180° - \theta_{inc}$ and $\phi_s = 0°$ for $\beta = 0°$ are accompanied by peaks of the $\hat{Z}_{11}$- and $\hat{Z}_{22}$-element. For tilt angle $\beta = 50°$ or $\beta = 90°$ the $\hat{Z}_{11}$- and $\hat{Z}_{22}$-elements do not have peaks at that direction but only in the forward direction $\theta_s = \theta_{inc}$. The peak at $\theta_s = 180° - \theta_{inc}$ and $\phi_s = 0°$ for tilt angle $\beta = 0°$ coincides with the specular reflection direction of a plane. The results of Adams and Bettenhausen (2012) for the $\hat{Z}_{11}$- and the $\hat{Z}_{21}$- element for size parameter $x \approx 4$ fit qualitatively with the $\hat{Z}_{11}$- and the $\hat{Z}_{21}$-element for tilt angle $\beta = 0°$ in Fig. 9. Interestingly, the large plate aggregate in Fig. 10 with similar size parameter $x$ as the plate type 1 habit in Fig. 9 does not show these peaks. There is also no strong backscattering for nadir incidence direction. Fig. 10 shows at $671\,\text{GHz}$ and for several incidence angles $\theta_{inc}$ and tilt angles $\beta$ the upper left block of the normalized scattering matrix $\hat{\boldsymbol{Z}}_{\text{aro}}(\theta_{inc}, \theta_s, \phi_s)$ of large plate aggregate for size parameter $x \approx 3$. In contrast to the plate type 1 habit in Fig. 9, the $\hat{Z}_{21}$- and $\hat{Z}_{12}$-elements are practically zero. This means unpolarized incoming radiation scattered by the large plate aggregate does not show much polarization. On the other hand, at $167\,\text{GHz}$ the $\hat{Z}_{21}$- and $\hat{Z}_{12}$-elements are non-zero and significantly differ between the different tilt angles $\beta$. Fig. 11 shows the upper left block of the normalized scattering matrix $\hat{\boldsymbol{Z}}_{\text{aro}}(\theta_{inc}, \theta_s, \phi_s)$ of the same large plate aggregate as in Fig. 10 but at $167\,\text{GHz}$. At $167\,\text{GHz}$ the size parameter for this particle is $x \approx 0.75$. Compared to Fig. 10 the scattering is less focused toward the forward scattering direction.

The data from the simulated scattering matrix can be used for simulations of passive and active observations. However, for simulations of horizontally scanning radars the scattering matrix in the backscattering direction has to be handled with care.

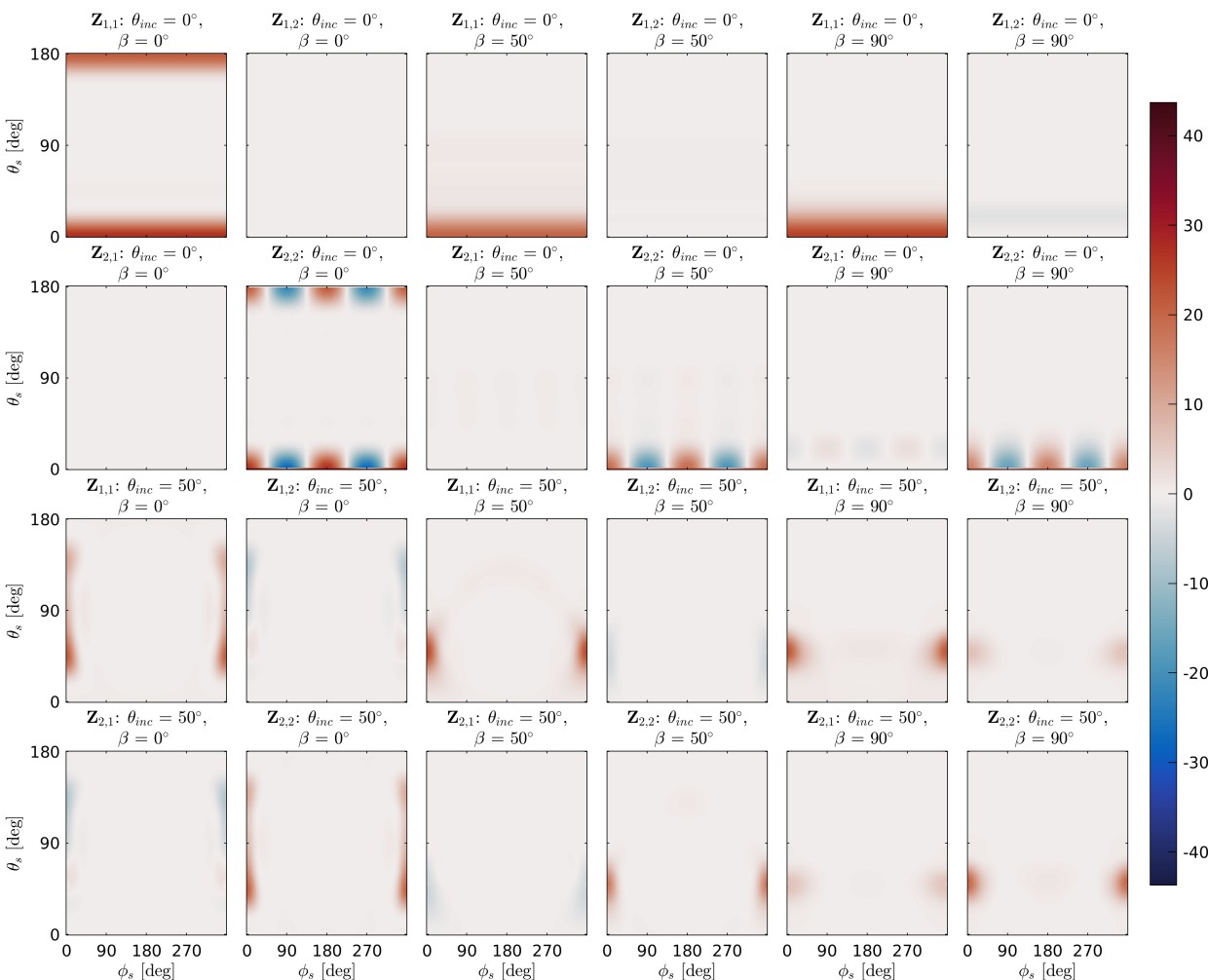

**Figure 9.** The upper left block of the normalized scattering matrix $\hat{\boldsymbol{Z}}$ of plate type 1 with a volume equivalent diameter of $429\,\mu m$ (Fig. 4) at $671\,\mathrm{GHz}$ as function of the polar scattering angle $\theta_s$ and the azimuth scattering angle $\phi_s$ for a set of tilt angles $\beta$ and incidence angles $\theta_{inc}$. A volume equivalent diameter of $429\,\mu m$ at $671\,\mathrm{GHz}$ corresponds to size parameter $x \approx 3$.

In the spherical harmonics representation of the Mueller matrix, the polarization at the poles, which are in the forward and backward direction, is not well represented. This can result in errors for the polarization. Most of this is averaged out due to the orientation averaging and the transformation to the scattering matrix, but there can be some residual effects for the polarization at the backscattering direction. This will be revised for the next iteration of the database.

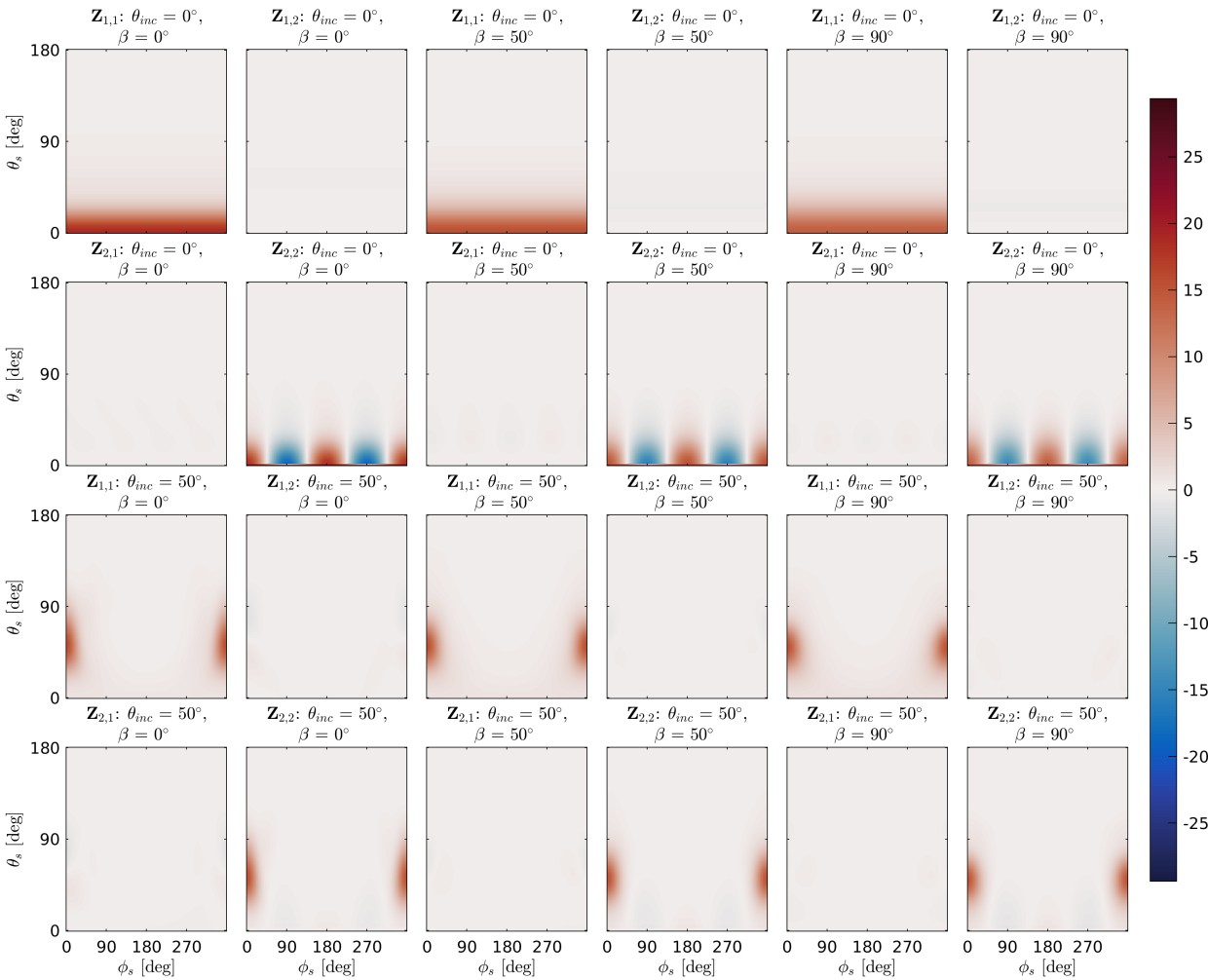

**Figure 10.** The upper left block of the normalized scattering matrix $\hat{\boldsymbol{Z}}$ of large plate aggregate with a volume equivalent diameter of $427\,\mu m$ (Fig. 4) at $671\,\mathrm{GHz}$ as function of the polar scattering angle $\theta_s$ and the azimuth scattering angle $\phi_s$ for a set of tilt angles $\beta$ and incidence angles $\theta_{inc}$. A volume equivalent diameter of $427\,\mu m$ at $671\,\mathrm{GHz}$ corresponds to size parameter $x \approx 3$.

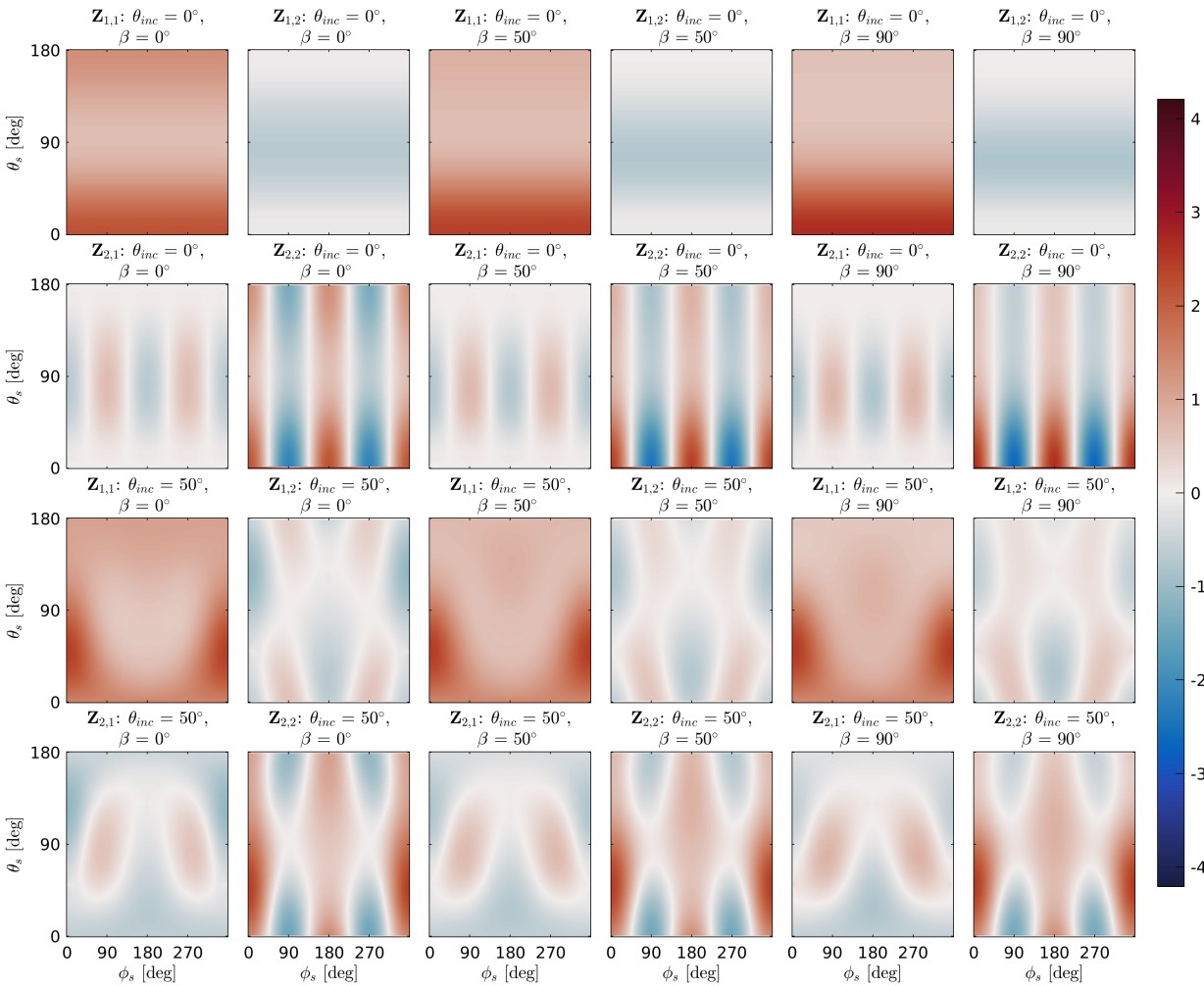

**Figure 11.** The upper left block of the normalized scattering matrix $\hat{\boldsymbol{Z}}$ of large plate aggregate with a volume equivalent diameter of $427\,\mu\text{m}$ (Fig. 4) at $167\,\text{GHz}$ as function of the polar scattering angle $\theta_s$ and the azimuth scattering angle $\phi_s$ for a set of tilt angles $\beta$ and incidence angles $\theta_{inc}$. A volume equivalent diameter of $427\,\mu\text{m}$ at $167\,\text{GHz}$ corresponds to size parameter $x \approx 0.75$.

 **6 Radiative transfer simulations**

In this section, we show radiative transfer simulations at $166\,\text{GHz}$ using azimuthally randomly oriented scatterers in order to give an example of the capabilities of the simulated scattering data. For the radiative transfer simulations, 200 atmospheric profiles over the tropical pacific were taken from one of the EarthCARE scenes. These scenes were prepared for the EarthCARE mission with Environment Canada's high-resolution numerical weather prediction model known as the Global Environmental Multiscale Model (GEM, Côté et al., 1998). The GEM scenes have a resolution of $250\,\text{m}$ and include two liquid hydrometeor species (liquid clouds and rain) and four frozen hydrometeor species (cloud ice, snow, graupel and hail). The profiles were randomly selected except for that they should cover the whole possible brightness temperature space as uniformly as possible.

The simulations were done using the Atmospheric Radiative Transfer Simulator (ARTS, Buehler et al., 2018; Eriksson et al., 2011) version 2.3.1118. The discrete ordinate iterative solver (DOIT, Emde, 2004) was used as scattering solver within ARTS. The simulations of Rayleigh–Jeans brightness temperatures were done using independent pixel approximation (IPA) with a local incidence angle of $49°$ for a satellite orbit height of $407\,\text{km}$ at $165.1\,\text{GHz}$ and $166.9\,\text{GHz}$, which were averaged to mimic the GMI's $166\,\text{GHz}$ channel. Within ARTS, gas absorption was taken into account by using the HITRAN data base (Rothman et al., 2013) and the MT_CKD model for the continuum absorption of water vapor and molecular nitrogen in version 2.52 (Mlawer et al., 2012). The gas absorption of molecular oxygen was processed by using the full absorption model of Rosenkranz (1998), modified by the values from Tretyakov et al. (2005). The ocean surface emissivity was calculated with the Tool to Estimate Sea-Surface Emissivity from Microwaves to sub-Millimeter waves (TESSEM2, Prigent et al. (2017)) implementation within ARTS, using the surface speed and temperature from the GEM profiles.

The Milbrandt-Yau two-moment microphysics (Milbrandt and Yau, 2005a, b) implementation within ARTS with the same hydrometeor types and size distributions as for the GEM runs was used. The Milbrandt-Yau two-moment microphysics assumes a modified gamma distribution (MGD) with characteristic parameters for each individual hydrometeor;

$$N(x) = N_0 x^\nu \exp\left(-\lambda x^\mu\right),\tag{29}$$

with the parameters $N_0$ and $\lambda$, which are functions of the number density and the hydrometeor content, and the parameters $\mu$ and $\nu$. The parameters $\mu$ and $\nu$ are fixed for each hydrometeor type and are summarized in Table 3. The Milbrandt-Yau two-moment bulk microphysics use the particle maximum diameter as independent variable $x$ for the size distribution.

The scattering properties for the hydrometeors were taken from Eriksson et al. (2018) except for cloud ice and snow. The database of Eriksson et al. (2018) contains among others the single scattering properties of hydrometeors that are modeled to be consistent with the m-D parameters of the Milbrandt-Yau two-moment bulk microphysics scheme. The particles inside the database of Eriksson et al. (2018) are assumed to be totally randomly oriented.

For cloud ice and snow the azimuthally randomly oriented plate type 1 and the azimuthally randomly oriented large plate aggregate are used, respectively. No averaging of the scattering data of the particles with its mirrored version was done for the radiative transfer simulation. Normally, this is done to assure that the scattering medium, in our case ice clouds, are mirror symmetric to the incidence plane. Mirror symmetric particles like the plate type 1 automatically fulfill this, but unsymmetric particles like the large plate aggregate generally do not. Due to the orientation averaging and the random structure of the large

**Table 3.** Size distribution parameters and the scatterer shape of the radiative transfer simulations. The size distribution parameters were taken from the source code of the Milbrandt-Yau two-moment bulk microphysics (Milbrandt and Yau, 2005a, b) of the GEM model. Except for cloud ice and snow the scattering properties were taken from Eriksson et al. (2018).

| | MGD parameter | | scatterer habits | | |
|---|---|---|---|---|---|
| | $\nu$ | $\mu$ | Fig. 12 | Fig. 13 | Fig. 15 |
| cloud water | 1 | 1 | Liquid Sphere, ID 25 | Liquid Sphere, ID 25 | Liquid Sphere, ID 25 |
| rain | 0 | 1 | Liquid Sphere, ID 25 | Liquid Sphere, ID 25 | Liquid Sphere, ID 25 |
| cloud ice | 0 | 1 | Plate Type 1 (ARO) | Plate Type 1 (ARO) | Plate Type 1 (ARO) |
| snow | 0 | 1 | Large plate aggr. (ARO) | Large plate aggr. (ARO) | Plate Type 1 (ARO) |
| graupel | 0 | 1 | GEM Graupel, ID 33 | - | GEM Graupel, ID 33 |
| hail | 0 | 1 | GEM Hail, ID 34 | GEM Hail, ID 34 | GEM Hail, ID 34 |

plate aggregate the effect of the non-mirror symmetry are so small that we neglected it for the radiative transfer simulations. For the simulations the azimuthally randomly oriented particles are orientation-averaged over Gaussian distributed $\beta$ angles with zero mean and increasing standard deviation. Six different orientation states were prepared for the simulations in order to mimic different stages of fluttering of the particle. Additionally, the azimuthally randomly oriented particles were averaged over uniformly distributed $\beta$ angle to show the results for total random orientation. The used single scattering properties are summarized in Table 3.

## 6.1 Results and discussion

Fig. 12 shows the vertical polarization of the brightness temperature $T_{bv}$ and the polarization difference $T_{bv} - T_{bh}$ as function of the frozen water path (FWP) for the different orientations. The FWP is the sum of each vertically integrated mass content of the four frozen hydrometeors. The plate type 1 habit for ice clouds and the large plate aggregate habit for snow were used for the simulation, see Table 3 for the other hydrometeors. The vertical polarization of the brightness temperature $T_{bv}$ decreases with increasing frozen water path from $\approx 280\,\mathrm{K}$ at a FWP of $\approx 10^{-2}\,\mathrm{kg\,m^{-2}}$ to $\approx 85\,\mathrm{K}$ at a FWP of $\approx 20\,\mathrm{kg\,m^{-2}}$. The polarization difference $T_{bv} - T_{bh}$ increases with increasing FWP till a maximum is reached at a FWP of $\approx 5\,\mathrm{kg\,m^{-2}}$ and then decreases with increasing FWP. The maximum of the polarization difference depends on the orientation state. For total horizontal orientation the maximum polarization difference is $\approx 11\,\mathrm{K}$. With increased standard deviation (fluttering) the maximum polarization difference decreases down to $\approx 2.5\,\mathrm{K}$ for totally randomly oriented particles. The orientation depending polarization difference also indicates that particle orientation is not only an issue for dual polarized observations but also for single polarized observations. Ignoring orientation can cause a negative bias for vertically polarized observations and in a positive bias for horizontally polarized observations.

Additionally, Fig. 12 shows the polarization difference $T_{bv} - T_{bh}$ as function of the vertical polarized brightness temperature $T_{bv}$. The polarization difference has a bell shaped distribution with a flat top and its maximum at $\approx 195\,\mathrm{K}$ for total horizontal orientation. With increased standard deviation the curve gets flatter. For small standard deviations ($\leq 10°$) the bell like dis-

tributions of the polarization difference are similar to the mean polarization differences that Gong and Wu (2017) estimated from GMI measurements over tropical ocean and the mean polarization differences that Defer et al. (2014) estimated from MADRAS. The results of Gong and Wu (2017) and of Defer et al. (2014) are additionally shown in Fig. 12 as gray solid and dashed lines, respectively. Though MADRAS has a slightly higher incidence angle than GMI and measures at $157\,\text{GHz}$ instead of $166\,\text{GHz}$, the observations of GMI and MADRAS are similar.

Additional tests show that the polarization difference and the brightness temperature are mainly influenced by snow and graupel. For these tests (not shown) one hydrometeor at a time was set to zero, while the others were unchanged, and the simulations for the 200 profiles and 7 orientation states were rerun. Cloud liquid and rain have an impact on single profiles but do not change the overall behavior of the polarization difference. The influence of ice clouds is negligible, because most of the ice cloud particles are too small to cause significant scattering at $166\,\text{GHz}$. Hail does not need to be considered, because within the 200 profiles its content is very little and therefore does not cause any significant scattering. Setting graupel or snow to zero strongly alters the polarization difference and the brightness temperature.

For the simulations shown in Fig. 13 the mass content and number density of graupel was added to snow but without changing the total amount of frozen water mass content. In this case snow is the only significant cause of scattering. Compared to Fig. 12 the minimum brightness temperature $T_{bv}$ is higher by $\approx 40\,\text{K}$, which means that the scattering of the large plate aggregate habit is weaker than the graupel habit. The reason is that the graupel habit, due to its higher density, has a larger scattering coefficient than the large plate aggregate. More interesting is how the polarization differs. The polarization difference $T_{bv} - T_{bh}$ distribution has indications of a bell like distribution but compared to Fig. 12 it does not reach zero for the minimum brightness temperature $T_{bv}$ and it is flatter. Furthermore, the polarization difference maximum is shifted by $\approx 30\,\text{K}$ to lower brightness temperature and is slightly higher. Down to $T_{bv} \approx 170\,\text{K}$ the polarization differences for small standard deviations ($\leq 10°$) are similar to the observed polarization differences of Gong and Wu (2017) and of Defer et al. (2014). For $T_{bv} \lesssim 170\,\text{K}$ the polarization differences are larger than the observed ones. Around brightness temperature $T_{bv} = 125\,\text{K}$, approximately the smallest simulated brightness temperature, the polarization difference is roughly twice as large as for the similar brightness temperature in Fig. 12 and the observations of Gong and Wu (2017) and of Defer et al. (2014).

The bell like distribution of the polarization difference $T_{bv} - T_{bh}$ in Fig. 13 is caused by two opposing effects. On one hand increasing the amount of scatterers results in increased scattering and increased polarization difference. On the other hand, increasing the amount of scatterers results in increased multi-scattering and decreased polarization difference. For a small amount of scattering the polarization increase dominates, while for a large amount of scattering polarization decrease dominates.

In Fig. 13 snow is the only significant cause of scattering, whereas in Fig. 12 snow and graupel are the causes of scattering. The smaller polarization differences in Fig. 12 compared to Fig. 13 for brightness temperatures $T_{bv} < 220\,\text{K}$ show that the composition of the scatterers, in addition to multi-scattering, reduces the polarization. As the amount of frozen particles increases the composition changes. For small amounts of frozen hydrometeors the amount of snow dominates, whereas the amount of graupel dominates for large amounts of frozen hydrometeors (see Fig. 14). Graupel is simulated by the GEM graupel habit of the database of Eriksson et al. (2018). Due to its total random orientation and its sphere-like shape, the GEM

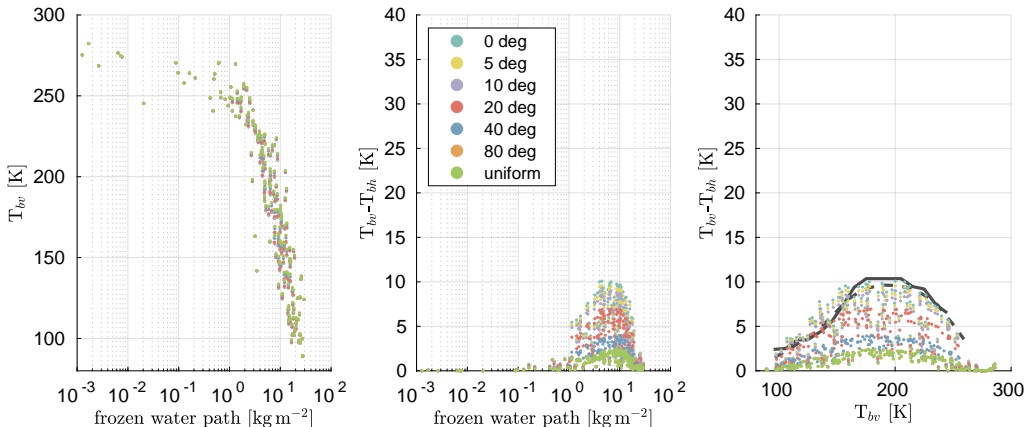

**Figure 12.** Simulated brightness temperature at $166\,\text{GHz}$ for 200 randomly selected atmospheric profiles. For each of these atmospheric profiles the scattering properties of the azimuthally randomly oriented scatterers are orientation averaged over 7 different distributed $\beta$ angles with zero mean and different standard deviation. The different colors denote the standard deviation of the $\beta$ angle distribution and the distribution type. For the used scatterers, see Table 3. The gray line solid line denotes the mean polarization difference over tropical ocean from GMI observations at $166\,\text{GHz}$ of Gong and Wu (2017) and the gray dashed line the mean polarization difference over tropical ocean from MADRAS observations at $157\,\text{GHz}$ of Defer et al. (2014).

graupel habit causes only negligible polarization at $166\,\text{GHz}$. For small amounts of frozen hydrometeors snow dominates the scattering and increasing the amount of frozen hydrometeors results in an increase in scattering and polarization difference. With increasing amounts of frozen hydrometeors multi-scattering and scattering by graupel increase. Both result in a decrease of the polarization difference. As a consequence, the polarization difference in Fig. 12 is smaller for $T_{bv} < 220\,\text{K}$ and the maximum polarization difference is at higher brightness temperatures than in Fig. 13.

As an additional scenario, the large plate aggregate habit for snow was replaced by the plate type 1 habit and the simulations for the 200 profiles and 7 orientation states were rerun, which is shown in Fig. 15. The polarization difference $T_{bv} - T_{bh}$ distribution has similar shape as in Fig. 12, but it has a roughly three times higher magnitude and a much higher spread. The brightness temperature $T_{bv}$ differs only slightly. This shows that the polarization difference not only depends on the orientation but also on the shape. For a standard deviation of $\approx 40°$ the bell like distribution of the polarization difference is comparable to the mean polarization differences of Gong and Wu (2017) and of Defer et al. (2014).

The comparison of the three different scenarios with the observations of Gong and Wu (2017) and of Defer et al. (2014) shows that snow simulated as large plate aggregate with small standard deviations ($\leq 10°$) or as plate type 1 with standard deviations in the order of $\mathcal{O}\,(40°)$ is compatible with the observations if additionally graupel is included within the simulations. Without graupel, the observed decrease of the polarization differences for brightness temperature $T_{bv} < 170\,\text{K}$ cannot be reached.

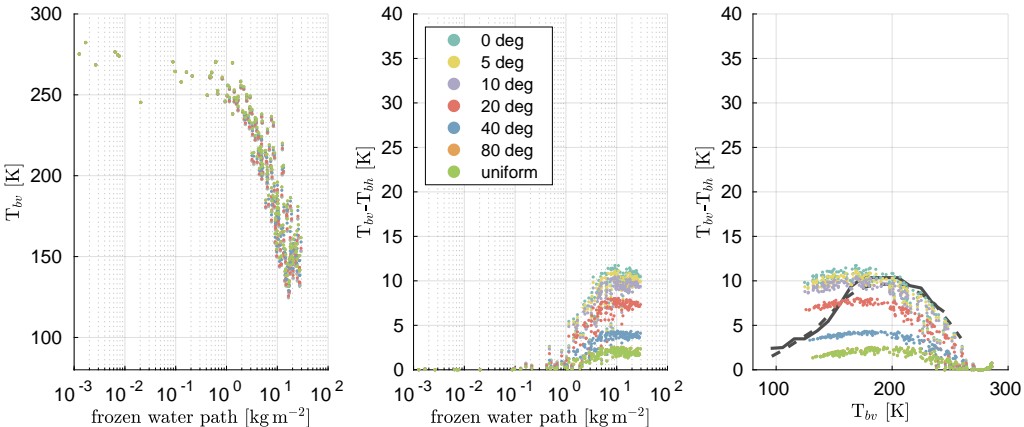

**Figure 13.** Same as Fig. 12 but the mass content and the number density of graupel added to snow.

## 7    Summary

We provide microwave and submillimeter wave scattering simulations of azimuthally randomly oriented ice crystals with a fixed but arbitrary tilt angle. For the simulations, DDA simulations made with ADDA were combined with a self developed orientation averaging approach. The scattering of 51 sizes of hexagonal plates (plate type 1) between $10\,\mu\text{m}$ and $2,596\,\mu\text{m}$ volume equivalent diameter and 18 sizes of hexagonal plate aggregates (large plate aggregate) between $197\,\mu\text{m}$ and $4,563\,\mu\text{m}$ for 35 frequencies between $1\,\text{GHz}$ and $864\,\text{GHz}$ and 3 temperatures ($190\,\text{K}$, $230\,\text{K}$ and $270\,\text{K}$) were simulated. The scattering data for azimuthal random orientation is much more complex than for total random orientation. For total random orientation the scattering matrix $\boldsymbol{Z}_{\text{tro}}(\Theta)$ depends only on one angle and the extinction matrix $\boldsymbol{K}_{tro}$ has no angular dependency at all and has only one independent entry. For azimuthal random orientation the scattering matrix $\boldsymbol{Z}_{\text{aro}}(\theta_{inc},\theta_s,\phi_s)$ depends on three angles and the extinction matrix $\boldsymbol{K}_{aro}(\theta_{inc})$ depends on the incidence angle and has three independent entries. Furthermore, the tilt angle $\beta$ increases the complexity. For a finite set of incidences and tilt angles in which the incidence angles $\theta_{inc}$ span a range from $0°$ to $180°$ with a $5°$ spacing and the tilt angles $\beta$ span a range from $0°$ to $90°$ for plate type 1 and from $0°$ to $180°$ for large plate aggregates with a $10°$ spacing the scattering data has a size of $181\,\text{GB}$. This is roughly 20 times bigger than the database of Eriksson et al. (2018). The scattering database of the azimuthally randomly oriented particles is publicly available from Zenodo (https://doi.org/10.5281/zenodo.3463003). It is organized so that the Python 3 interface of the database of Eriksson et al. (2018) can be used to extract and interact with the data.

To give an example of the capabilities of the dataset, we conducted radiative transfer simulations of polarized GMI measurements of differently fluttering ice crystals at $166\,\text{GHz}$. The radiative transfer simulations were conducted using ARTS (Buehler et al., 2018; Eriksson et al., 2011) and assuming Milbrandt-Yau two-moment microphysics (Milbrandt and Yau, 2005a, b) with two liquid hydrometeor species (rain, liquid clouds) and four frozen hydrometeor species (cloud ice, snow, graupel, and hail). For slightly fluttering snow and ice particles, the simulations show polarization differences up to $11\,\text{K}$ using the azimuthally

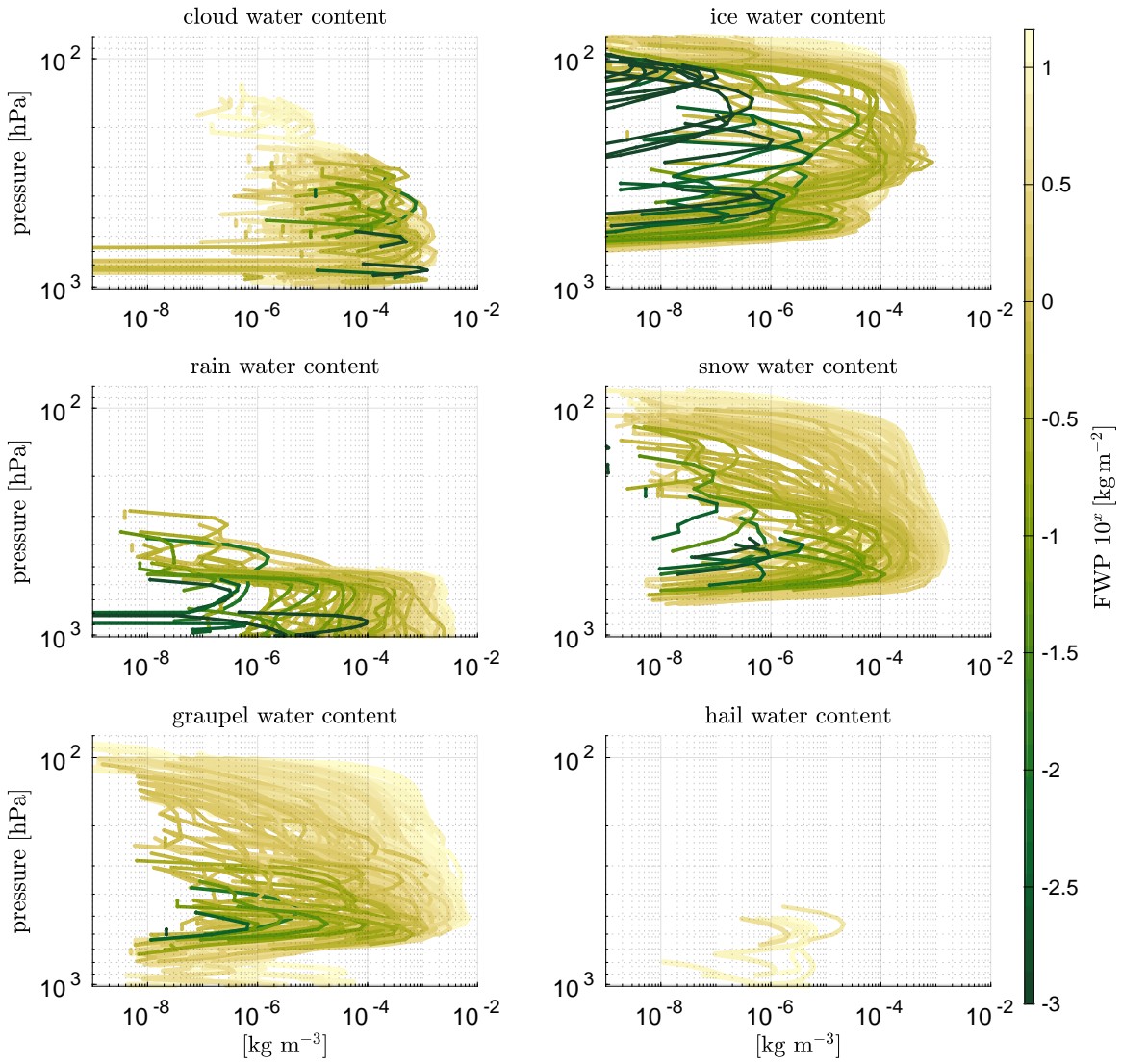

**Figure 14.** Hydrometeor content profiles used for the radiative transfer simulation in Fig. 12. The color indicates the frozen water path (FWP) of each atmospheric profile.

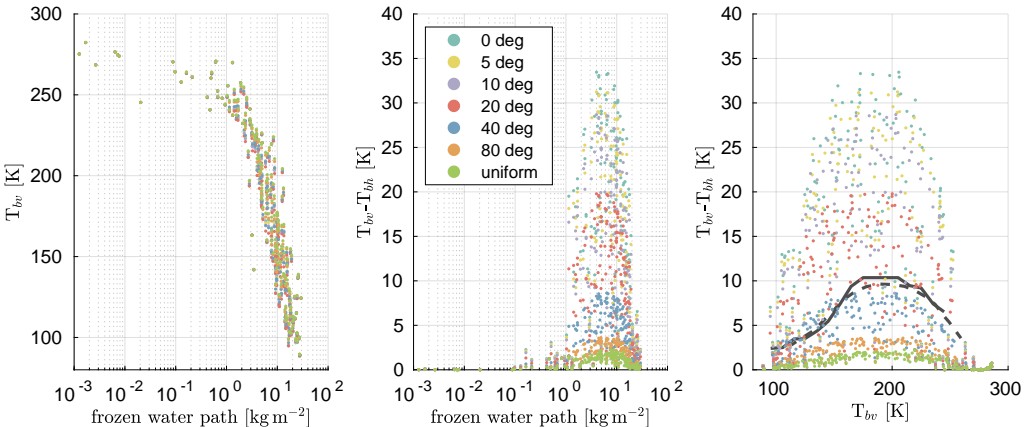

**Figure 15.** Same as Fig. 12 but with plate type 1 for snow instead of large plate aggregate.

randomly oriented large plate aggregate habit for snow, the plate type 1 habit for cloud ice and totally oriented particles for the other four hydrometeors. The simulations cover the observed brightness temperatures and polarization differences from Gong
and Wu (2017) and Defer et al. (2014). Further analysis shows that not only multi-scattering affects the polarization but also the hydrometeor composition. The polarization difference and the brightness temperature are mainly influenced by snow and graupel. Exchanging the large plate aggregate habit with the plate type 1 habit for snow results in roughly three times bigger polarization difference. For strongly fluttering snow and ice particles, the simulations using the plate type 1 habit for snow and ice are similar to Gong and Wu (2017) and Defer et al. (2014). Particle orientation also affects single polarized observa-
tions. Ignoring orientation can cause a negative bias for vertically polarized observations and in a positive bias for horizontally polarized observations.

Using the new scattering data, retrievals of polarized observations from GMI, MADRAS and especially the upcoming ICI can give us new insights for the understanding of clouds. For example, to the authors' knowledge none of the latest atmospheric weather and climate models handle orientation. Furthermore, polarization can give us additional information on the shape of
the particle.

*Data availability.*  The scattering database of the azimuthally randomly oriented particles is publicly available from Zenodo (https://doi.org/ 10.5281/zenodo.3463003). The data of the radiative transfer simulations of Sect. 6 is also publicly available from Zenodo (https://doi.org/10. 5281/zenodo.3475897). The data from the DDA simulations in truncated spherical harmonics representation is available upon request from us.

*Competing interests.*  The authors declare that they have no conflict of interest.

*Author contributions.* Manfred Brath has developed the orientation averaging approach, set up and conducted the scattering and the radiative transfer simulations and written the article's text. Robin Ekelund prepared the scatterer shape data, designed the database structure and contributed text. Patrick Eriksson has acted as project leader, initiated the database and suggested the averaging approach. Patrick Eriksson and Stefan A. Buehler participated in planning of the database and have contributed text. Oliver Lemke helped to set up and to conduct the scattering simulations and prepared the data for publication.

*Acknowledgements.* A large part of this work was produced inside a study funded by EUMETSAT (Contract No. EUM/COS/LET/16/879389). The study manager at EUMETSAT was Christophe Accadia, who provided appreciated feedback and inspiration. Robin Ekelund and Patrick Eriksson were further supported financially by the Swedish National Space Agency (SNSA) under grant 150/14. The authors would like to thank Howard Barker from Environment and Climate Change Canada for providing the GEM model simulations. Stefan A. Buehler was supported by the Deutsche Forschungsgemeinschaft (DFG, German Research Foundation) under Germany's Excellence Strategy — EXC 2037 "Climate, Climatic Change, and Society" — Project Number: 390683824, contributing to the Center for Earth System Research and Sustainability (CEN) of Universität Hamburg. Finally, our thanks to the ARTS radiative transfer community for their help with using ARTS.

## Appendix A: Initial particle alignment

Before any orientation averaging can be performed, the initial orientation of the particle has to be defined. The alignment algorithm is mainly based on aligning the principal moments of inertia axes along the Cartesian coordinate axes. Also, a number of special cases are treated in order to make the alignment consistent between particles and not dependent on small numerical differences. The result of the algorithm is that the particle fulfills the following criteria: the principal axis of the particle with the largest inertia is aligned along the z-axis, and its principal axis with the smallest inertia along the x-axis.

The algorithm involves a several steps. For particles that possess no symmetries, one step can be skipped. The algorithm operates on a coordinate grid and consists of the following steps:

1. First, the particle mass center coordinate $\overline{r}$ is calculated, according to

$$\overline{r} = \sum_{i=1}^{N} m_i r_i, \tag{A1}$$

where $r_i$ is (3x1) column vector describing the coordinate of the grid point with index $i$, and $m_i$ is the mass of the corresponding dipole. The dipole grid is then displaced so that the mass center is located at the origin.

2. Next, the inertia matrix $I$ relative to the origin is calculated using

$$I = -\sum_{i=1}^{N} m_i [R]_i^2, \tag{A2}$$

where $[\boldsymbol{R}]_i$ is the skew-symmetric matrix associated with coordinate $\boldsymbol{r}$, defined as

$$[\boldsymbol{R}] = \begin{pmatrix} 0 & -z & y \\ z & 0 & -x \\ -y & x & 0 \end{pmatrix}. \tag{A3}$$

$\boldsymbol{I}$ contains the products of inertia along the Cartesian coordinate axes, i.e.

$$\boldsymbol{I} = \begin{pmatrix} I_{xx} & I_{xy} & I_{xz} \\ I_{xy} & I_{yy} & I_{yz} \\ I_{xz} & I_{yz} & I_{zz} \end{pmatrix}. \tag{A4}$$

Since $\boldsymbol{I}$ is real and symmetric, it can be diagonalized using eigenvector decomposition, as

$$\boldsymbol{\Lambda} = \boldsymbol{Q}\boldsymbol{I}\boldsymbol{Q}^T, \tag{A5}$$

where $\boldsymbol{\Lambda}$ is a diagonal matrix with elements $I_1$, $I_2$ and $I_3$, which are called the principal moments of inertia. The
diagonalization is performed in such way that $I_1 \leq I_2 \leq I_3$. The columns of $\boldsymbol{Q}$, $Q_1$, $Q_2$ and $Q_3$, are the corresponding
principal axes.

It follows that $\boldsymbol{Q}$ is a rotation matrix, which rotates the $x$, $y$ and $z$-axes to corresponding axes of inertia. Thus, to align the
particle principal axes to the coordinate axes, one has to rotate the particle grid by the inverse of $\boldsymbol{Q}$, i.e. $\boldsymbol{Q}^T$. In order to
ensure that the rotation does not mirror the particle (that the rotation is pure), one has to make sure that $\det\left(\boldsymbol{Q}^T\right) = 1$.
The rotation matrix $\boldsymbol{A}$ is thus calculated as

$$\boldsymbol{A} = \frac{\boldsymbol{Q}^T}{|\boldsymbol{Q}^T|}. \tag{A6}$$

After the rotation, recalculation of the inertia matrix should yield

$$\boldsymbol{I} = \begin{pmatrix} I_{xx} & 0 & 0 \\ 0 & I_{yy} & 0 \\ 0 & 0 & I_{zz} \end{pmatrix}, \tag{A7}$$

With

$$I_{xx} \leq I_{yy} \leq I_{zz}. \tag{A8}$$

This criteria must always be satisfied, i.e. any of the remaining steps must make sure that it does not violate the condition.

3. If the particle contains symmetries, then two or all of the principal moments of inertia can be equal. This means that
the rotation in the previous step is unambiguous, i.e. several possible orientations fulfill Eq. A8. As an example, for
hexagonal plates, $I_{xx} = I_{yy}$, meaning that its orientation in the xy-plane is unambiguous. It is desirable to remove this
uncertainty, which here is done by minimizing the particle dimensions along the coordinate axes. Three cases are possible
and are treated as follows:

- $I_{xx} = I_{yy} = I_{zz}$: The particle is spherically symmetric (for example, a six bullet rosette), hence no rotation will have an impact on $\boldsymbol{I}$. First, the particle dimension along the z-axis is minimized by rotation around the x and y-axis. Similarly, the particle dimension along the x-axis is then maximized by rotation around the z-axis.

- $I_{yy} = I_{zz}$: The particle is symmetric around the x-axis (a hexagonal column for example). The particle dimension along the z-axis is minimized by rotation around the x-axis.

- $I_{yy} = I_{xx}$: The particle is symmetric around the z-axis (for example, a hexagonal plate). The particle dimension along the x-axis is maximized by rotation around the z-axis

4. In the final step, it is determined whether the particle is aligned upside down or upright. First, the minimum circumsphere of the particle is calculated, with its corresponding center. If the center is found to be below the mass-center of the particle (with respect to the z-axis), then the particle is said to be aligned upright. Vice versa, it is said to be aligned upside down in the case when the sphere center is above the mass center. In this case, the particle is rotated 180° around the x-axis to be upright.

## Appendix B: Particle rotation

The key point in our averaging approach is the rotation of the particle for the averaging process. When rotating the particle the wave reference system rotates as well. The wave reference system is the coordinate system that is defined by the incidence direction and the particle system. The changed direction $\hat{e}_{i,rot}$ for a desired orientation is given by

$$\hat{e}_{i,rot} = \boldsymbol{R}_{\alpha\beta\gamma}\hat{e}_i \tag{B1}$$

with $\hat{e}_i$ the non-rotated incidence or scattering direction and $\boldsymbol{R}_{\alpha\beta\gamma}$ the rotation matrix. The rotation matrix $\boldsymbol{R}_{\alpha\beta\gamma}$ is

$$\boldsymbol{R}_{\alpha\beta\gamma} = \boldsymbol{R}_{\alpha}\left(\alpha\right)\boldsymbol{R}_{\beta}\left(\beta\right)\boldsymbol{R}_{\gamma}\left(\gamma\right) = \begin{pmatrix} R_{11} & R_{12} & R_{13} \\ R_{21} & R_{22} & R_{23} \\ R_{31} & R_{32} & R_{33} \end{pmatrix} \tag{B2}$$

with the Euler angles $\alpha$, $\beta$, and $\gamma$. The rotation matrix elements $R_{ij}$ are

$$R_{11} = \cos(\gamma)\cos(\beta)\cos(\alpha) - \sin(\gamma)\sin(\alpha) \tag{B3}$$

$$R_{12} = \cos(\gamma)\cos(\beta)\sin(\alpha) + \sin(\gamma)\cos(\alpha) \tag{B4}$$

$$R_{13} = -\cos(\gamma)\sin(\beta) \tag{B5}$$

$$R_{21} = -\sin(\gamma)\cos(\beta)\cos(\alpha) - \cos(\gamma)\sin(\alpha) \tag{B6}$$

$$R_{22} = -\sin(\gamma)\cos(\beta)\sin(\alpha) + \cos(\gamma)\cos(\alpha) \tag{B7}$$

$$R_{23} = \sin(\gamma)\sin(\beta) \tag{B8}$$

$$R_{31} = \sin(\beta)\cos(\alpha) \tag{B9}$$


$$R_{32} = \sin(\beta)\sin(\alpha) \tag{B10}$$

$$R_{33} = \cos(\beta) \tag{B11}$$

with Euler angles $\alpha$, $\beta$, and $\gamma$ (Tsang et al., 2000). When the wave reference system changes, the polarization directions change,
too. The polarization directions of each simulated Mueller matrix and extinction matrix are relative to the wave reference system, which is different for each incidence angle. This means the original polarization directions of the Mueller matrix and the extinction matrices change under rotation as indicated in Fig. B1. The rotation about the laboratory Z-axis by the Euler angle $\alpha$ does not change the polarization, because the vertical polarization direction stays always in the plane spanned by incidence direction unit vector $\hat{e}_{ki}$ and the laboratory z-axis and the horizontal polarization direction stays parallel to the x-
y-plane. But the combined rotations by the Euler angles $\beta$ and $\gamma$ do change. After the combined rotation the original vertical polarization unit vector $\hat{e}_v$ is rotated out of the plane spanned by incidence direction unit vector $\hat{e}_{ki}$ and the laboratory z-axis by angle $\varphi$ and original horizontal polarization unit vector $\hat{e}_h$ is rotated out of the x-y-plane by angle $\varphi$. After the rotation using $\boldsymbol{R}_{\alpha\beta\gamma}$ the polarization of the Mueller matrix $\boldsymbol{M}$ and the extinction matrix $\boldsymbol{K}$ need to be transformed to the laboratory polarization using the stokes rotation matrix $\boldsymbol{L}$ (Mishchenko et al., 2002)

$$\boldsymbol{L}(\varphi) = \begin{pmatrix} 1 & 0 & 0 & 0 \\ 0 & \cos 2\varphi & -\sin 2\varphi & 0 \\ 0 & \sin 2\varphi & \cos 2\varphi & 0 \\ 0 & 0 & 0 & 1 \end{pmatrix}. \tag{B12}$$

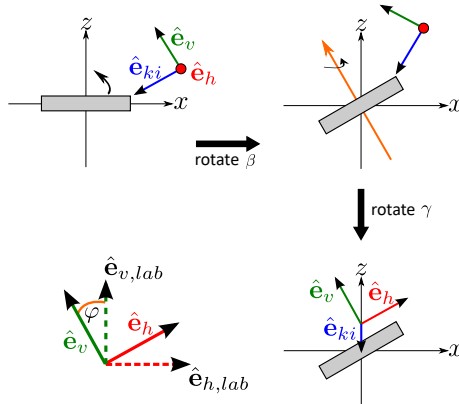

**Figure B1.** Change of the polarization directions under rotation. (top left) the incidence direction unit vector $\hat{e}_{ki}$ together with the vertical polarization unit vector $\hat{e}_v$ and the horizontal polarization unit vector $\hat{e}_h$, which are fixed to the particle, before the rotation is performed. (top right) the unit vectors after the rotation by angle $\beta$ and (bottom right) after the rotation by angle $\gamma$. As indicated (bottom left) the polarization vectors after the rotation by angles $\beta$ and $\gamma$ are twisted by angle $\varphi$ compared to the laboratory unit vectors.

The Mueller matrix $\boldsymbol{M}_{rot}$ and the extinction matrix $\boldsymbol{K}_{rot}$ of the rotated particle are given by

$$\boldsymbol{M}_{rot} = R^*_{\alpha\beta\gamma}(\boldsymbol{M}) = \boldsymbol{L}(\varphi)\,\boldsymbol{M}\left(\boldsymbol{R}_{\alpha\beta\gamma}(\theta_{inc}, \phi_{inc}), \boldsymbol{R}_{\alpha\beta\gamma}(\theta'_s, \phi'_s)\right)\boldsymbol{L}(-\varphi) \tag{B13}$$

and

$$\boldsymbol{K}_{rot} = R^*_{\alpha\beta\gamma}(\boldsymbol{K}) = \boldsymbol{L}(\varphi)\,\boldsymbol{K}\left(\boldsymbol{R}_{\alpha\beta\gamma}(\theta_{inc}, \phi_{inc})\right)\boldsymbol{L}(-\varphi)\,. \tag{B14}$$

The rotation angle $\varphi$ is

$$\varphi = \mathrm{atan2}\left(\hat{e}_v \cdot \hat{e}_{h,lab},\, \hat{e}_v \cdot \hat{e}_{v,lab}\right) \tag{B15}$$

with the rotated vertical polarization direction $\hat{e}_v$ , the horizontal polarization direction in the laboratory system

$$\hat{e}_{h,lab} = \hat{e}_{v,lab} \times \hat{e}_{ki}\,, \tag{B16}$$

the vertical polarization direction in the laboratory system

$$\hat{e}_{v,lab} = (\hat{e}_z \times \hat{e}_{ki}) \times \hat{e}_{ki}\,, \tag{B17}$$

and z-direction $\hat{e}_z$.

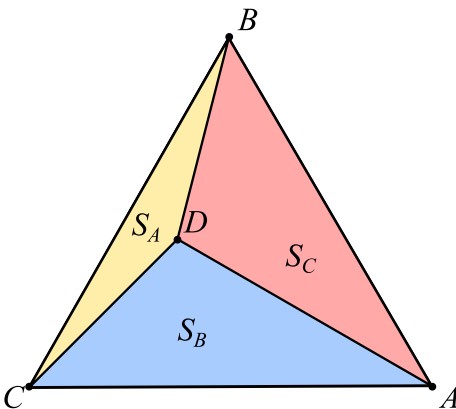

**Figure C1.** Geometry of triangular barycentric interpolation.

### Appendix C: Barycentric interpolation

On an icosahedral grid, any arbitrary point on the sphere is accompanied by three nearest points that form a equilateral triangle. Within this triangle the value at that point can be interpolated from the vertices of the triangle. An illustration of the problem is shown in Fig. C1. The vertices $A$, $B$, and $C$ form the equilateral triangle $ABC$. The point $D$ is the evaluation point. Two vertices and evaluation point $D$ form a sub-triangle. For example, the vertices $B$ and $C$ and the evaluation point $D$ form the triangle $BCD$ on the opposing side of vertex $A$. The idea behind the barycentric interpolation is to use the ratio of the area of a sub-triangle and the area of the triangle $ABC$ as interpolation weights. The weight belonging to vertex $A$ is

$$w_A = \frac{S_A}{S_{ABC}} \tag{C1}$$

with $S_A$ the area of sub-triangle $BCD$ and $S_{ABC}$ the area of the triangle $ABC$. The weights belonging to the other two vertices are analogue to the weight of vertex $A$. The area $S$ of a triangle is using Heron's formula

$$S_i = \sqrt{s\left(s-u\right)\left(s-v\right)\left(s-w\right)} \tag{C2}$$

with

$$s = \frac{u+v+w}{2} \tag{C3}$$

and $u$, $v$, $w$ the sides of the triangle $i$. The interpolated value $f_{int}$ at the evaluation point $D$ is

$$f_{int}\left(D\right) = w_A f\left(A\right) + w_B f\left(B\right) + w_C f\left(C\right) \tag{C4}$$

with $f\left(i\right)$ the value at a vertex $i$.

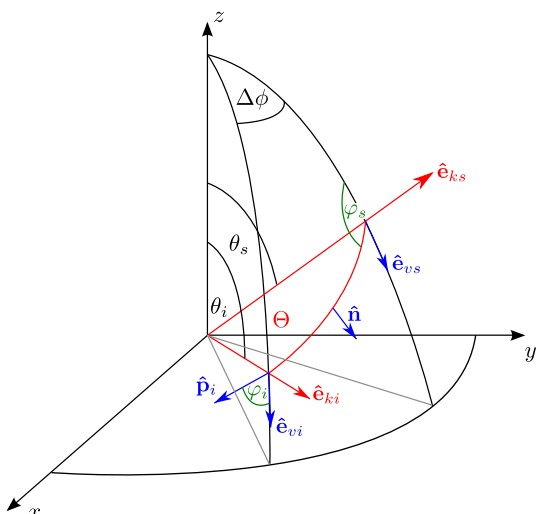

**Figure D1.** Scattering geometry in the laboratory system

## Appendix D: Transformation of the averaged Mueller matrix to the averaged scattering matrix

Between the scattering matrix averaged $\boldsymbol{Z}$ and the averaged Mueller matrix $\boldsymbol{M}$ the following relationship holds

$$\boldsymbol{Z}\left(\theta_{inc},\theta_s,\phi_s,\beta\right) = \frac{1}{k^2}\boldsymbol{L}\left(-\varphi_s\right)\boldsymbol{M}\left(\theta_{inc},R\left(\theta_s',\phi_s'\right),\beta\right)\boldsymbol{L}\left(\varphi_i\right) \tag{D1}$$

with $k$ the angular wavenumber, $\boldsymbol{L}$ the Stokes rotation matrix (Eq. B12), $\varphi_i$ and $\varphi_s$ the polarization rotation angles, and $R\left(\theta_s',\phi_s'\right)$ the rotation operator that transforms the incidence direction related coordinate system to the laboratory system.

As defined in Sect. 2.2, the incidence azimuth direction is zero. In that case the incidence direction vector is always within the X-Z-plane. The rotation operator $R\left(\theta_s',\phi_s'\right)$ then is

$$\begin{pmatrix} \theta_s \\ \phi_s \end{pmatrix} = R\begin{pmatrix} \theta_s' \\ \phi_s' \end{pmatrix} = \begin{pmatrix} \arccos\left(-\sin\theta_{inc}\sin\theta_s'\cos\phi_s' + \cos\theta_{inc}\cos\theta_s'\right) \\ \mathrm{atan2}\left(\sin\theta_s'\sin\phi_s',\cos\theta_{inc}\sin\theta_s'\cos\phi_s' + \sin\theta_{inc}\cos\theta_s'\right) \end{pmatrix}. \tag{D2}$$

The Stokes rotation matrices $\boldsymbol{L}\left(-\varphi_s\right)$, $\boldsymbol{L}\left(\varphi_i\right)$ transform the polarization basis from relative to the scattering direction to relative to incidence direction. Fig. D1 shows the geometry for polarization basis transformation. The Stokes rotation matrix $\boldsymbol{L}\left(-\varphi_s\right)$ describes the rotation by angle $\varphi_s$. This is the angle between the scattering plane and the plane that is spanned by the unit vector of the scattering direction $\hat{\boldsymbol{e}}_{ks}$ and the laboratory Z-axis. The scattering plane is the plane that is spanned by the unit vector of the incidence direction $\hat{\boldsymbol{e}}_{ki}$ and the unit vector of the scattering direction $\hat{\boldsymbol{e}}_{ks}$. The Stokes rotation matrix $\boldsymbol{L}\left(\varphi_i\right)$ describes the rotation by angle $\varphi_i$. This is the angle between the scattering plane and the plane that is spanned by the unit vector of the incidence direction and the laboratory Z-axis. The unit vector $\hat{\boldsymbol{e}}_{kj}$ describing the incidence or scattering direction

is

$$\hat{e}_{kj} = \begin{pmatrix} \sin\theta_j \cos\phi_j \\ \sin\theta_j \sin\phi_j \\ \cos\theta_j \end{pmatrix} \tag{D3}$$

and the unit vector of the vertical polarization $\hat{e}_{vj}$ for the incidence direction or the scattering direction is

$$\hat{e}_{vj} = \begin{pmatrix} \cos\theta_j \cos\phi_j \\ \cos\theta_j \sin\phi_j \\ -\sin\theta_j \end{pmatrix} \tag{D4}$$

with $j = i, s$ for the incidence direction and the scattering direction, respectively. The rotation angle is

$$\varphi_j = \begin{cases} -\arccos(\hat{e}_{vj} \cdot \hat{p}_j) & , \hat{e}_{vj} \cdot \hat{n}_j \geq 0 \\ \arccos(\hat{e}_{vj} \cdot \hat{p}_j) & , \hat{e}_{vj} \cdot \hat{n}_j < 0 \end{cases} . \tag{D5}$$

with the unit vector

$\hat{p}_j = \hat{n} \times \hat{e}_{kj}$ $\qquad\qquad$ (D6)

that is parallel to scattering plane and orthogonal to $\hat{e}_{kj}$. The normal vector

$$\hat{n} = \frac{\hat{e}_{ks} \times \hat{e}_{ki}}{\sin\Theta} \tag{D7}$$

is orthogonal to the scattering plane. The scattering angle $\Theta$, which is the angle between the incidence direction and the scattering direction is

$\sin\Theta = |\hat{e}_{ks} \times \hat{e}_{ki}|$ $\qquad\qquad$ (D8)

In the actual implementation each matrix element $M_{ij,aro}(\theta_{inc}, \theta'_s, \phi'_s)$ of the averaged Mueller matrix is represented as a spherical harmonics series over the scattering directions $\theta'_s, \phi'_s$. For the calculation of the averaged scattering matrix $Z_{aro}$, the Mueller matrix elements $M_{ij,aro}(\theta_{inc}, \theta'_s, \phi'_s)$ in angular grid representation are used. The resulting scattering matrix elements $Z_{ij,aro}$ in angular grid representation are expanded afterwards as spherical harmonics series over the scattering angles $\theta_s$ and

$\phi_{s_s}$.

## Appendix E: Spherical harmonics expansion of the Mueller and scattering matrix elements

Each matrix element $X_{ij}(\theta_{inc}, \phi_{inc}, \theta_s, \phi_s)$ of the Mueller matrix or the scattering matrix is expanded in a spherical harmonics series over the scattering directions $(\theta_s, \phi_s)$.

$$X_{ij}(\theta_{inc}, \phi_{inc}, \theta_s, \phi_s) = \sum_{l=0}^{l_{max}} \sum_{m=-l}^{l} C_{lm}(\theta_{inc}, \phi_{inc}) Y_{lm}(\theta_s, \phi_s) \tag{E1}$$

with $Y_{lm}$ the spherical harmonic function of the $l$-th and $m$-th order and with

$$C_{lm}\left(\theta_{inc},\phi_{inc}\right) = \int_{\Omega_s} X_{ij}\left(\theta_{inc},\phi_{inc},\theta_s,\phi_s\right) Y_{lm}^*\left(\theta_s,\phi_s\right) d\Omega_s \tag{E2}$$

the expansion coefficients of the incidence direction $(\theta_{inc},\phi_{inc})$. To save data space, the expansion of $X_{ij}$ is truncated to the value $l_{max}$. $l_{max}$ is defined as the lowest $l$ for which it holds that

$$\left[\int_{\Omega_s}\left|X_{ij}\left(\theta_{inc},\phi_{inc},\theta_s,\phi_s\right) - \sum_{l=0}^{l_{max}}\sum_{m=-l}^{l} C_{lm}\left(\theta_{inc},\phi_{inc}\right) Y_{lm}\left(\theta_s,\phi_s\right)\right|^2 d\Omega_s\right]^{\frac{1}{2}} < \varepsilon_{M11}. \tag{E3}$$

$\varepsilon_{M11}$ is $0.5\%$ of the standard deviation over the scattering directions $(\theta_s,\phi_s)$ of the $X_{11}\left(\theta_{inc},\phi_{inc}\right)$ matrix element. For the actual calculation of the spherical harmonics, the SHTns library version 2.8 (Schaeffer, 2013) and its Python interface are used.

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
