# Peer review of "Microwave and submillimeter wave scattering of oriented ice particles"

_Atmospheric Measurement Techniques, 2019_

## Referee Comment (RC1) · Anonymous Referee #1 · 3 Dec 2019

General comments

This article by Brath et al. presents a novel, highly valuable study and database of the properties of oriented snow particles in the atmosphere at low to high microwave frequencies. This has been a goal of the local research community for years, and the reviewer is rather glad that he came across it for review.

The resulting database is gigantic, and the complexities of assembling this database are discussed at length throughout the manuscript. Great care was taken to describe all of the conventions and equations involved in the scattering calculations, particle rotations and subsequent radiative transfer simulations. The reviewer congratulates the authors on this achievement.

[Figure]

It is quite possible, however, for readers to become lost in this level of detail and lose the main thrust of the paper. Also, earlier sections of the manuscript (e.g. pp. 3 and 10) refer the reader to details in section 4 (p. 11). As such, the reviewer suggests that the authors attempt to simplify by moving some of these details into appendices and somewhat reordering the manuscript. There are also many small points (both scientific and formatting) that should be addressed. The overall recommendation is to revise and resubmit.

Section-by-section comments

Abstract:

The abstract is too vague. It states that you performed simulations and made a database for use with the upcoming Ice Cloud Imager. Your summary section contains information that should be emphasized here. Results from Sections 6 and 7 can further provide examples of why undertaking the construction of this database was worthwhile.

- Line 7: "The additional tilt angle adds an additional dimension" -> . . . adds an additional degree of freedom. Dimension can be rather confusing in the context of this paper.

- Line 8: dipol -> dipole

- Line 9: Perhaps mention that these habits were first introduced in a previous paper. Mention that the database covers multiple temperatures.

- Line 10: The data is -> The data are

Introduction:

You need information on why polarized scattering properties are important. What new information content would they provide for data assimilation / forecasting? Metop-SG-B's ICI instrument (launching in the early 2020s) will need better models of snow parti-

cle scattering to properly retrieve ice cloud properties. To provide for this, you need a few key components: accurate ice particle shapes, a polarized radiative transfer model, accurate orientation distributions, and polarization-sensitive dataset of ice particle scattering. Eriksson et al. (2018) provides the shapes, ARTS provides the radiation model, you assume the orientation distributions and generate the scattering dataset that studies / people / instruments can use.

- Line 15: "channels of these passive" -> "channels of passive"

- Lines 17 and elsewhere: "Currently, ... GPM and MADRAS ... are the only spaceborne microwave radiometer that measure polarization at ice cloud frequencies. GPM and MADRAS observe polarization around 160 GHz."

MADRAS was declared non-operational about two years after launch, and it is no longer collecting scientifically valid data. The sentence should reflect that.

- Line 19: You might want to discuss the abundance of polarized data available at around 90 GHz. Polarized measurements are also available on Metop-C and on GCOM-W1, but are strongly affected by surface contamination.

- Line 22: The mentioning of particle orientations is rather abrupt. You need a few expository sentences here.

- Line 24: "realistically shape" -> "realistically shaped"

- Line 24: "that also possess an orientation" – This dangles from the end of the sentence, and should be rephrased.

- Line 28: "one orientations" -> "one orientation"

Particle orientation:

This section can get rather technical, and so it is important for the reader to be guided through possible misunderstandings.

[Figure]

- Lines 49 and elsewhere: "spherical symmetry" is a bit confusing, and you seem to be using two competing meanings of this term throughout the manuscript.

Consider a symmetric 6-bullet rosette.

In Appendix A, line 592, spherical symmetry occurs when Ixx = Iyy = Izz.

Contrast to line 49: "If the particle possesses spherical symmetry there is no particle orientation, because it does not matter from which side the particle with spherical symmetry is viewed or how it is rotated – it will always look the same.": This seems more like radial symmetry.

- Line 50: Last sentence of paragraph is cumbersome. How about "The particles considered in this paper are not radially symmetric and may be oriented."?

- Lines 52-62: "In general, the orientation of a particle in a three dimensional space can be described by a set of three parameters. The three Euler angles are one such parameter set." - You need to assert in this section that your choice of rotation angles are not necessarily the same rotation angles used elsewhere. There are six pure Euler angle schemes (intrinsic rotations), six Tait-Bryan conventions (extrinsic rotations; some literature sources also consider these to be Euler angles), and several mixed approaches.

- Line 52: "three dimensional" -> "three-dimensional"

- Lines 56,57 and in many places elsewhere: Something went wrong with the PDF rendering of some of the symbols used in your manuscript (e.g. zyz' notation is displayed as z[box]z'). This happens on different machines (macs, Windows) and using different PDF readers.

- Line 62: "important to know" -> "important to note"

- Line 63: "Additionally to the Euler angles" -> "In addition to the Euler angles"

- Line 66: You are considering only generally oblate particles (and your particle model

is discussed later in the paper). You mention plates here, but it is good to explicitly state that you assume only oblate shapes. If you have something more prolate-shaped (i.e. columns), then its general alignment to vertical or horizontal becomes a very complicated function of drag and other local conditions (you can get preferential vertical instead of horizontal alignment).

- Lines 80, 81 and 98: Define the abbreviations (TRO, ARO) in lines 80,81 instead of in a subsequent figure caption (line 98).

- Line 90: Total scattering angle Θ is a function of the angle between incoming and outgoing direction, and it might be useful to include the equation here.

- Lines 94+, and 70-79: You seem to be aligning your ensemble of particles in the same way, regardless of the different moments of inertia, sizes and aspect ratios of the particles. Why not allow for different particles in your ensemble to have different preferential alignments, perhaps using von Mises-Fisher or Fisher-Bingham-Kent distributions? It's worth discussing, especially since related work has been presented by the GPM team.

This also relates to my comment in section 5 – can the raw (per-orientation) data be made available for users to manipulate independently?

- Lines 100-116: The paragraph is wordy and would be hard to understand for someone outside of the immediate field. Lines 101 and 111 state, "to get a better picture of it" and "to get a better idea of it". You might need to add in a descriptive figure here.

Basic setup and shape data

This section reads well.

- Line 118: Amsterdam DDA's name was changed. See a recent version of the manual for their rationale.

- Lines 125-126: As described elsewhere in the manuscript, the two hydrometeor habits have multiple shapes in each habit. The text here is somewhat misleading and should

be rephrased. Or, prefix with the sentences in lines 127-128.

- Line 134: Volume equivalent diameter should be defined. It has multiple meanings in the field, and I am assuming that you mean the diameter of an equal-volume sphere made of solid ice, later used in line 310.

- Line 134: Same with maximum diameter. Assuming you mean in three dimensions.

- Line 135: Why are the sizes slightly different?

- Line 142: Why are the frequencies slightly different?

- Page 7: 886 GHz is quite high! What interdipole spacing did you use when calculating these results?

- Line 158: Spacing. "Fig. 4 b" -> "Fig. 4b" to match "Fig. 4a" on line 156.

- Line 158: "This approach is analogue to the analytic T-matrix method, only in a much more numerical way." I am uncertain if many readers will appreciate the analogy.

- Line 168: "stokes" -> "Stokes"

- Line 195: The number of incidence angles seems to be rather low. The reviewer recognizes that adding more would be prohibitively expensive, and that the manuscript is already a substantial improvement on what was available before. However, it might be worth commenting on in the text.

- Line 202: "appendix" -> "Appendix"

- Section 4.1: Particle rotation: No comments here.

- Lines 252, 253, 256: "stokes" -> "Stokes"

Results of the scattering simulations

There were 69 particles overall, and 7245 cases, with over a million core hours, and about 1.5 TB or raw data. However, are users of the database are restricted to the

orientationally-averaged set? The summary section, line 517, implies that only the summarized data are available.

- Line 290: spacing. "scattering matrix Zaroand" -> "Zaro and"

- Eqn 33: there is a spurious dot between the two lines of the equation. Was this supposed to be a comma?

- Line 295: Wrong font for absorption vector "a".

- Line 307: "3" -> "three"

- Lines 337-350, 375, Fig. 8,9 captions: PDF rendering problem with the asymmetry parameter

- Line 361: "Eqn." -> "Eq." to match how you abbreviate everywhere else.

Radiative transfer simulations

No major comments.

- Line 456: "addionally" -> "additionally"

- Line 487: "sphere like" -> "sphere-like"

Summary

Good section overall. Some of the information here should be highlighted in the abstract.

- Line 552: fix opening quote before Climate

Appendices

No major comments.

- Lines 569, 577, 614, 616, 617: fix rendering

- Line 592: See comment in Particle Orientation section.

References

Various formatting typos.

- Line 666: "in: 2016" -> "in 2016"?

- Line 670: "157?GHz"

- Line 700: "Iet"?

- Lines 701-704: Title capitalization is inconsistent with other references.

- Line 706: Cambridge University Press (capitalization)

- Line 726: "ADDA: Capabilities". Capitalization in contrast to line 662.

---

## Short Comment (SC1) · 3 Dec 2019

The paper presents an important step forward in the currently available scattering databases of snow particles at microwave frequencies by assuming the possibility of ice particles with preferential orientations. This is an important contribution which I recommend for publication, but I would also like to list some comments aiming to improve the value of the paper.

1. The orientation averaging technique lacks some validation. A very basic sanity check would be to calculate the integral over $\cos(\beta)$ at the various $\theta_{inc}$ and compare with the previously published database (DB) for total random orientation (TRO). Another useful plot to include would be the convergence of the integral

with respect to the number of points of the icosahedral grid. At line 195 it is stated that a variable number of points is used (between 162 and 2562), perhaps these convergence plots would clarify why, sometimes, a smaller number of orientation samples is sufficient.

2. The averaging scheme is presented as a solution to various challenges that sequentially appear in the text. It is hard, sometimes, to follow this approach because it requires to rethink about the setup many times without a clear final goal to aim to. I want to suggest to introduce the three main reference frames of the problem from the beginning: these are the laboratory (satellite) reference frame, the particle reference frame, and the wave reference frame. By doing so, one can states from the beginning that the scope is to have the polarized scattering properties defined with respect to the satellite reference frame and some transformations are needed because for scattering calculations the wave reference frame is a more natural option used in scattering codes. Also what is called the orientation of the non-rotated particle is nothing less than the particle reference frame.

3. Line 62. This phrase, somehow implies that there is a special subset of rotation matrices that are orthogonal and no couple of rotation matrices are commutative with respect to multiplication. I think all rotation matrices are orthogonal and some rotation matrices do commute (the ones around the same axis).

4. Line 87. For TRO $p_\beta$ should be $\frac{1}{2}$ and $\beta$ should be uniformly distributed in terms of $\cos(\beta)$. Otherwise, the integral does not compute to 1 when K=1

5. Line 110. I think here the non-symmetry is respect to the scattered azimuth, not the incident which is actually irrelevant for $Z_{aro}$.

6. Line 121. ADDA can actually also compute scattering properties for distributions of angles through input files, this includes azimuthally averaging. The reason

why this is not used in the study is that this approach involves the solution of the computationally demanding DDA problem for slightly different orientations many times (for the different combinations of tilt angle and wave incidence).

7. Line 130. $D_0$ should have explicit units, which I assume are $\mu m$.

8. Line 179-182. I do not see why a regular grid is advantageous for resolving the for/back-ward scattering peaks. A regular grid means that the azimuth and polar angles are equally spaced. The points at the same polar angle are getting closer in azimuth distance as the polar angle approaches the poles. The scattering peaks mean that there is a high variability of the scattering intensity with respect to the polar angle and thus would demand an increased resolution in polar angles. The polar angle resolution is always the same here.

9. Lines 209-214. In my opinion, two points are missing in the list of steps: first is the projection over spherical harmonics of the scattered fields. And the second is the barycentric interpolation of the gridded data. The second is important because it clarifies that the computed properties for a certain $\beta$ and $\theta_i$ are actually coming from slightly different angles.

10. Line 220. The three rotation matrices are different. Perhaps a better notation would be $R_{\alpha\beta\gamma} = R_z(\alpha)R_y(\beta)R_z(\gamma)$

11. Line 284. What is called accuracy $\epsilon$=1% I think is the internal stopping criterion for the ADDA iterative solver and should not be confused with the accuracy of the calculations which is hard to evaluate and yet not clearly understood. Perhaps the authors should include in the supplementary material, for just one particle and one orientation what is the effect on the scattering properties (just plot phase functions ) of this choice of $\epsilon$ with respect to the default value of $10^{-5}$ (three orders of magnitude smaller!).

12. Line 381. In the figure, I see $\beta = 0$, 50, 90 but in the text, $\beta{=}30$ is mentioned, perhaps there is a typo?

13. Line 397-402. Here the authors state that the database is not optimized for radar calculations because the spherical harmonics projection is not good at forward and backward scattering. Perhaps the authors should better describe what they meant at line 177 with RMSE of less than 0.5% due to the spherical harmonics. 0.5% is actually quite insignificant for radar applications. Also this problem can be immediately solved by making available the original DDA computations at single orientations, perhaps by request to the corresponding author. I think this last piece would also make the paper fully compliant with the Copernicus open-data policy.

14. The scattering properties of hexagonal crystals are symmetric with respect to $\theta_i$ due to the planar symmetry of the particles. This is not true for aggregates that are not symmetric. The authors have oriented the aggregates according to their principal axis of inertia. This is, in general, a good fast approach, but it introduces an arbitrary decision about what is the direction of the main (vertical) axis of inertia. In my opinion, there is no clear criterion to decide whether this axis should look up or down. As a consequence, one could argue that the scattering properties for $\theta_i = \lambda$ should be averaged with those for $\theta_i = 180 - \lambda$ giving planar symmetry also to the aggregates and reducing the storage footprint of the database.

15. Equations (20) and (21) show how to rotate the polarization vectors of the Mueller matrix. I wonder if this is done before the barycentric interpolation. In my view, the scattering properties of the three vertexes should be first aligned with the direction and polarization of point D. If the forward/backward scattering direction lies within the triangle ABC this can cause quite dramatic cancellation due to the flipping of the polarization direction among the points A, B, and C.

---

## Referee Comment (RC2) · Anonymous Referee #2 · 9 Dec 2019

The manuscript "Microwave and submillimeter wave scattering of oriented ice particles" is well-written, logically constructed, and highly impactful. Databases of such oriented particles, particularly with complete phase and extinction matrix information, are not available, so this dataset is expected to be groundbreaking for microwave, millimeter-wave, and submillimeter-wave sensor modeling applications. The radiative transfer results are very encouraging, and show that the authors have done a good job of creating a useful dataset. After addressing some minor clarifying issues, this manuscript is ready for publication.

My biggest concern is the precision to which these calculations have been run (see lines 176-178; 283-285). I understand that these are computationally-expensive calculations, so improving on these numbers is beyond the scope of this paper. However,

the cross-polarization terms, i.e., Z12 and Z21, are orders of magnitude smaller than Z11, so these terms may be unreliable, and looking at the processed data, there seems to be a lot of noise that is of the same order of magnitude as the signal. Luckily these terms are small, and the largest expected contribution would be to radar polarimetric variables, especially LDR. I think the authors should make a note of this when discussion the precision relative to Z11 (and the other phase matrix terms).

The authors should make clear that the amplitude scattering matrix (equation 11) operates on the complex electric field terms.

The authors should explicitly state that orientational averaging must be done incoherently, that is at the the Mueller (or Phase) matrix stage, due to the power terms in the top left block of the Phase matrix.

When discussing mirror partners and mirror symmetry, please cite van de Hulst (1957) and Mishchenko (2002). There is a really nice explanation with stick figures in both publications.

Also in reference to mirror particles, for the RT simulations in section 6.1, were the particles averaged with their mirror partners (with respect to the incidence plane)? This is important for properly conditioning the Z12 and Z21 Phase matrix parameters for the target medium of preferential alignment with zero mean canting angle.

Technical Corrections:

There are minor typos throughout the manuscript that need to be fixed, but the document as a whole is very clear.

There are a few symbols that did not render properly, one of which was the asymmetry parameter.

---

## Author Comment (AC1) · 31 Jan 2020

**Answers to Interactive comment on "Microwave andsubmillimeter wave scattering of oriented ice particles" by Manfred Brath et al.**
**Anonymous Referee #1**

January 31, 2020

General comments

Reviewer:

**This article by Brath et al. presents a novel, highly valuable study and database of the properties of oriented snow particles in the atmosphere at low to high microwave frequencies. This has been a goal of the local research community for years, and the reviewer is rather glad that he came across it for review.**

**The resulting database is gigantic, and the complexities of assembling this database are discussed at length throughout the manuscript. Great care was taken to describe all of the conventions and equations involved in the scattering calculations, particle rotations and subsequent radiative transfer simulations.**

**The reviewer congratulates the authors on this achievement.**

**It is quite possible, however, for readers to become lost in this level of detail and lose the main thrust of the paper. Also, earlier sections of the manuscript (e.g. pp. 3 and 10) refer the reader to details in section 4 (p. 11). As such, the reviewer suggests that the authors attempt to simplify by moving some of these details into appendices and somewhat reordering the manuscript. There are also many small points (both scientific and formatting) that should be addressed. The overall recommendation is to revise and resubmit.**

Answer:

As suggested, we shifted the details of section 4 to the Appendix. Furthermore, we revised the paper considering the comments of all three reviewers.

Section-by-section comments

Abstract

Reviewer:

**The abstract is too vague. It states that you performed simulations and made adatabase for use with the upcoming Ice Cloud Imager. Your summary section con-tains information that should be emphasized here. Results from Sections 6 and 7 can further provide examples of why undertaking the construction of this database wasworthwhile.**

Answer:

We revised accordingly.

Reviewer:

**- Line 7: "The additional tilt angle adds an additional dimension" -> . . . adds an additional degree of freedom. Dimension can be rather confusing in the context of this paper.**

Answer:

We changed it. We now write: "The additional tilt angle further increases the complexity."

Reviewer:

**- Line 8: dipol -> dipole**

Answer:

Changed as suggested.

Reviewer:

**- Line 9: Perhaps mention that these habits were first introduced in a previous paper. Mention that the database covers multiple temperatures.**

Answer:

We now mention that the database covers multiple temperatures.

Reviewer:

**- Line 10: The data is -> The data are**

Answer:

We change it.

Introduction:

Reviewer:

**You need information on why polarized scattering properties are important. What new information content would they provide for data assimilation / forecasting? Metop-SG-B's ICI instrument (launching in the early 2020s) will need better models of snow particle scattering to properly retrieve ice cloud properties. To provide for this, you need a few key components: accurate ice particle shapes, a polarized radiative transfer model, accurate orientation distributions, and polarization-sensitive dataset of ice particle scattering. Eriksson et al. (2018) provides the shapes, ARTS provides the radiation model, you assume the orientation distributions and generate the scattering dataset that studies / people / instruments can use.**

Answer:

We revised the intoduction according to that. We now state, why polarization is important and why it is important to have scattering properties of oriented and realistically shaped particles. Furthermore, we rephrased the goal of the study and the idea behind the database.

Reviewer:

**- Line 15: "channels of these passive" -> "channels of passive"**

Answer:

Changed as suggested.

Reviewer:

**- Lines 17 and elsewhere: "Currently, . . . GPM and MADRAS . . . are the only spaceborne microwave radiometer that measure polarization at ice cloud frequencies. GPM and MADRAS observe polarization around 160 GHz." MADRAS was declared non-operational about two years after launch, and it is no longer collecting scientifically valid data. The sentence should reflect that.**

Answer:

We corrected that and mention now that due to mechanical failure MADRAS measured only till January 2013.

Reviewer:

**- Line 19: You might want to discuss the abundance of polarized data available at around 90 GHz. Polarized measurements are also available on Metop-C and on GCOM-W1, but are strongly affected by surface contamination.**

Answer:

We added some sentences about it. We now mention that there are polarized observations below $100\,GHz$. We further added that due to the low frequency, the sensitivity considering ice clouds is low Buehler et al.(2007), though there still can be enough sensitivity for precipitating ice, and that at theses frequencies surface contamination is an issue.

Reviewer:

**- Line 22: The mentioning of particle orientations is rather abrupt. You need a few expository sentences here.**

Answer:

Done as suggested. We restructured the introduction and gave some additional background considering polarization, see also answer to your first introduction comment.

Reviewer:

**- Line 24: "realistically shape" -> "realistically shaped"**

Answer:

Changed as suggested.

Reviewer:

**- Line 24: "that also possess an orientation" – This dangles from the end of the sentence, and should be rephrased.**

Answer:

Removed, due to restructuring of the introduction.

Reviewer:

**- Line 28: "one orientations" -> "one orientation"**

Answer:

Changed as suggested.

Particle orientation:

Reviewer:

**This section can get rather technical, and so it is important for the reader to be guided through possible misunderstandings.**

**- Lines 49 and elsewhere: "spherical symmetry" is a bit confusing, and you seem to be using two competing meanings of this term throughout the manuscript. Consider a symmetric 6-bullet rosette. In Appendix A, line 592, spherical symmetry occurs when Ixx = Iyy = Izz. Contrast to line 49: "If the particle possesses spherical symmetry there is no particle orientation, because it does not matter from which side the particle with spherical symmetry is viewed or how it is rotated – it will always look the same.": This seems more like radial symmetry.**

Answer:

According to the American Meteorological Society Glossary of Meteorology radial symmetry and spherical symmetry is the same in three dimensions. "Radial symmetry in two dimensions is often called circular symmetry; in three dimensions, spherical symmetry." http://glossary.ametsoc.org/wiki/Radial_symmetry Therefore, we did not change it.

Reviewer:

**- Line 50: Last sentence of paragraph is cumbersome. How about "The particles considered in this paper are not radially symmetric and may be oriented."?**

Answer:

We changed it to "The particles considered in this paper are not spherically symmetric and therefore can be oriented."

Reviewer:

**- Lines 52-62: "In general, the orientation of a particle in a three dimensional space can be described by a set of three parameters. The three Euler angles are one such parameter set." - You need to assert in this section that your choice of rotation angles are not necessarily the same rotation angles used elsewhere. There are six pure Euler angle schemes (intrinsic rotations), six Tait-Bryan conventions (extrinsic rotations; some literature sources also consider these to be Euler angles), and several mixed approaches.**

Answer:

We now state that there is no unique set of parameters and that there are different sets of them depending on the definition of the rotation axes.

Reviewer:

**- Line 52: "three dimensional" -> "three-dimensional"**

Answer:

Changed as suggested.

Reviewer:

**- Lines 56,57 and in many places elsewhere: Something went wrong with the PDF rendering of some of the symbols used in your manuscript (e.g. zyz' notation is**

**displayed as z[box]z'). This happens on different machines (macs, Windows) and using different PDF readers.**

Answer:

Unfortunately, there were some problems with the font. This happened when the manuscript was uploaded to AMT. We are aware of this.

Reviewer:

**- Line 62: "important to know" -> "important to note"**

Answer:

Changed as suggested.

Reviewer:

**- Line 63: "Additionally to the Euler angles" -> "In addition to the Euler angles"**

Answer:

Changed as suggested.

[Figure]

Reviewer:

**- Line 66: You are considering only generally oblate particles (and your particle model is discussed later in the paper). You mention plates here, but it is good to explicitly state that you assume only oblate shapes. If you have something more prolate-shaped (i.e. columns), then its general alignment to vertical or horizontal becomes a very complicated function of drag and other local conditions (you can get preferential vertical instead of horizontal alignment).**

Answer:

We revised that part and now discuss the validity of our assumption.

Reviewer:

**- Lines 80, 81 and 98: Define the abbreviations (TRO, ARO) in lines 80,81 instead of in a subsequent figure caption (line 98).**

Answer:

Done as suggested.

Reviewer:

**- Line 90: Total scattering angle is a function of the angle between incoming and outgoing direction, and it might be useful to include the equation here.**

Answer:

Done as suggested.

Reviewer:

**- Lines 94+, and 70-79: You seem to be aligning your ensemble of particles in the same way, regardless of the different moments of inertia, sizes and aspect ratios of the particles. Why not allow for different particles in your ensemble to have different preferential alignments, perhaps using von Mises-Fisher or Fisher-Bingham-Kent distributions? It's worth discussing, especially since related work has been presented by the GPM team. This also relates to my comment in section 5 – can the raw (per-orientation) data be made available for users to manipulate independently?**

Answer:

Considering your comment and after reading the specific lines again, the lines 70 - 79 may be misleading, because it was not stated clearly in manuscript what is the basic idea behind the database and its usage, which we now do. Our main assumption is that there is no preferred orientation of the particles in azimuth direction. Based on that we want the users to decide, which tilt angle $\beta$ (zenith orientation) or tilt angle distribution they need. That is why we calculate the scattering properties for several tilt angles $\beta$ for all particle sizes and shapes.

In the revised introductions, we now state: "The idea behind the scattering database is that the users can use scattering data of a desired zenith orientation or combine the data of different zenith orientations to mimic any desired distribution of zenith orientations." We also rephrased the goal of the study, see answers to section-by-section

comments for introduction.

Reviewer:

**- Lines 100-116: The paragraph is wordy and would be hard to understand for someone outside of the immediate field. Lines 101 and 111 state, "to get a better picture of it" and "to get a better idea of it". You might need to add in a descriptive figure here.**

Answer:

Done as suggested.

Basic setup and shape data

Reviewer:

**This section reads well.**
**- Line 118: Amsterdam DDA's name was changed. See a recent version of the manual for their rationale.**

Answer:

Changed as suggested.

[Figure]

Reviewer:

**- Lines 125-126: As described elsewhere in the manuscript, the two hydrometeor habits have multiple shapes in each habit. The text here is somewhat misleading and should be rephrased. Or, prefix with the sentences in lines 127-128.**

Answer:

We reordered it as suggested and further revised that section considering your and the other reviewers' comments.

Reviewer:

**- Line 134: Volume equivalent diameter should be defined. It has multiple meanings in the field, and I am assuming that you mean the diameter of an equal-volume sphere made of solid ice, later used in line 310.**

Answer:

We added the definition of the volume equivalent diameter to the text. The volume equivalent diameter is defined as the diameter of a solid ice sphere with the same mass as the particle.

Reviewer:

**- Line 134: Same with maximum diameter. Assuming you mean in three dimensions.**

Answer:

We added the definition of the maximum diameter to the text. The maximum diameter is defined as the diameter of the minimum circumscribed sphere of a particle.

Reviewer:

**- Line 135: Why are the sizes slightly different?**

Answer:

The plate type 1 habit in our study has slightly different sizes than the plate type 1 in Eriksson et al. (2018), because an older version of shape data was used than in Eriksson et al. (2018) and given the high computational costs of the scattering calculation a recalculation was not feasible. We added previous sentence to the text.

Reviewer:

**- Line 142: Why are the frequencies slightly different?**

Answer:

Due to a rounding mistake when the simulation was set up, the frequencies of the plate type 1 habit slightly deviate from the frequencies of the large plate aggregate habit by at maximum $0.5\,GHz$. We added previous sentence to the text.

Reviewer:

**- Page 7: 886 GHz is quite high! What interdipole spacing did you use when calculating these results?**

Answer:

For all particles considered in our study holds

$$|m|\, kd < \frac{1}{2} \tag{1}$$

with $m$ the refractive index of ice, $k$ the wavelength and $d$ the dipole size. With the microwave refractive index of ice this result in roughly $22$ dipoles per wavelength. Furthermore, all simulated particles consist of at least $1,000$ dipoles so that small particles are reasonable resolved. We added a similar statement to Section Basic setup and shape data.

Reviewer:

**- Line 158: Spacing. "Fig. 4 b" -> "Fig. 4b" to match "Fig. 4a" on line 156.**

Answer:

Changed so that they now match.

Reviewer:

**- Line 158: "This approach is analogue to the analytic T-matrix method, only in a much more numerical way." I am uncertain if many readers will appreciate the analogy.**

Answer:

We removed it.

Reviewer:

**- Line 168: "stokes" -> "Stokes"**

Answer:

Changed as suggested.

Reviewer:

**- Line 195: The number of incidence angles seems to be rather low. The reviewer recognizes that adding more would be prohibitively expensive, and that the manuscript is already a substantial improvement on what was available before. However, it might be worth commenting on in the text.**

Answer:

We agree that it can seem to be, but the number of incidence angles is not low. Unfortunately, we forgot to mention in the text what accuracy we aim for the database. We now state in Section Basic setup and shape data that we aim for an accuracy of the scattering database in the order of a few percent. Relative to this, the number of incidence angles is sufficient. We further added some sentences considering the number of incidence angles.

Reviewer:

**- Line 202: "appendix" -> "Appendix"**

Answer:

Changed as suggested.

Reviewer:

**- Section 4.1: Particle rotation: No comments here.**
**- Lines 252, 253, 256: "stokes" -> "Stokes"**

Answer:

Changed as suggested.

Results of the scattering simulations

Reviewer:

**There were 69 particles overall, and 7245 cases, with over a million core hours, and about 1.5 TB or raw data. However, are users of the database are restricted to the orientationally-averaged set? The summary section, line 517, implies that only the summarized data are available.**

Answer:

Yes, only the averaged data is publicly available, because it is not feasible for us to host the non-averaged data, but the data can be given to anyone who is interested by contacting us. We added a similar statement to the text.

Reviewer:

**- Line 290: spacing. "scattering matrix Zaroand" -> "Zaro and"**

Answer:

Changed as suggested.

Reviewer:

**- Eqn 33: there is a spurious dot between the two lines of the equation. Was this supposed to be a comma?**

Answer:

We corrected it.

Reviewer:

**- Line 295: Wrong font for absorption vector "a".**

Answer:

We corrected it.

Reviewer:

**- Line 307: "3" -> "three"**

Answer:

Changed as suggested.

Reviewer:

**- Lines 337-350, 375, Fig. 8,9 captions: PDF rendering problem with the asymmetry parameter.**

[Figure]

Answer:

We corrected it, see answers to section-by-section Particle orientation.

Reviewer:

**- Line 361: "Eqn." -> "Eq." to match how you abbreviate everywhere else.**

Answer:

Changed as suggested.

Radiative transfer simulations

Reviewer:

**No major comments.**
**- Line 456: "addionally" -> "additionally"**

Answer:

Changed as suggested.

Reviewer:

**- Line 487: "sphere like" -> "sphere-like"**

Answer:

Changed as suggested.

Summary

Reviewer:

**Good section overall. Some of the information here should be highlighted in the abstract.**

Answer:

Done as suggested.

Reviewer:

**- Line 552: fix opening quote before Climate**

Answer:

Done as suggested.

Appendices

Reviewer:

**No major comments.**
**- Lines 569, 577, 614, 616, 617: fix rendering**

Answer:

We corrected it, see answers to section-by-section Particle orientation.

Reviewer:

**- Line 592: See comment in Particle Orientation section.**

Answer:

See first answer in answers to section-by-section Particle orientation.

References

Reviewer:

**Various formatting typos.**
**- Line 666: "in: 2016" -> "in 2016"?**

Answer:

No, it is correct as it is. It is the official citation from IEEE.

Reviewer:

**- Line 670: "157?GHz"**

Answer:

Corrected it.

Reviewer:

**- Line 700: "Iet"?**

Answer:

Changed to"IET".

Reviewer:

**- Lines 701-704: Title capitalization is inconsistent with other references.**

Answer:

Corrected it.

Interactive
comment
Reviewer:

**- Line 706: Cambridge University Press (capitalization)**

Answer:

Corrected it.

Reviewer:

**- Line 726: "ADDA: Capabilities". Capitalization in contrast to line 662.**

Answer:

Corrected it.

References

Buehler, S. A., Jiménez, C., Evans, K. F., Eriksson, P., Rydberg, B., Heymsfield, A. J., Stubenrauch, C. J., Lohmann, U., Emde, C., John, V. O., and et al.: A concept for a satellite mission to measure cloud ice water path, ice particle size, and cloud altitude, Quarterly Journal of the Royal Meteorological Society, 133, 109128, https://doi.org/ 10.1002/qj.143, 2007.

Eriksson, P., Ekelund, R., Mendrok, J., Brath, M., Lemke, O., and Buehler, S. A.: A general database of hydrometeor single scattering properties at microwave and sub-millimetre wavelengths, Earth System Science Data, 10, 1301-1326, https://doi.org/ 10.5194/essd-10-1301-2018, URL https://www.earth-syst-sci-data.net/10/1301/ 2018/,

2018.

---

## Author Comment (AC2) · 31 Jan 2020

**Answers to Interactive comment on "Microwave andsubmillimeter wave scattering of oriented ice particles" by Manfred Brath et al.**
**Anonymous Referee #2**

January 31, 2020

Reviewer:

**The manuscript "Microwave and submillimeter wave scattering of oriented ice particles" is well-written, logically constructed, and highly impactful. Databases of such oriented particles, particularly with complete phase and extinction matrix information, are not available, so this dataset is expected to be groundbreaking for microwave, millimeterwave, and submillimeter-wave sensor modeling applications. The radiative transfer results are very encouraging, and show that the authors have done a good job of creating a useful dataset. After addressing some minor clarifying issues, this manuscript is ready for publication.**

**My biggest concern is the precision to which these calculations have been run (see lines 176-178; 283-285). I understand that these are computationally-**

**expensive calculations, so improving on these numbers is beyond the scope of this paper. However, the cross-polarization terms, i.e., Z12 and Z21, are orders of magnitude smaller than Z11, so these terms may be unreliable, and looking at the processed data, there seems to be a lot of noise that is of the same order of magnitude as the signal. Luckily these terms are small, and the largest expected contribution would be to radar polarimetric variables, especially LDR. I think the authors should make a note of this when discussion the precision relative to Z11 (and the other phase matrix terms).**

Answer:

We have to admit that we forgot to address the accuracy of the database within the text. In the revised version, we do. Due to the high demands in view of computation time and the amount of data, we had to compromise in terms of the accuracy of the resulting scattering data, which we forgot to mention. Considering the measurement errors of existing and upcoming passive MW and SubMM sensors, which are in the order of $\mathcal{O}\left(1\,K\right)$, and the brightness temperature depression due to scattering of frozen hydrometeors, which is typically $<\ 100\,K$, we aim for an accuracy of the scattering database in the order of a few percent. We added a similar statement to the Section Basic setup and shape data and added it to the summary to clearly address this. Furthermore, we now relate the truncation of the spherical harmonics in Section Scattering calculations to the desired accuracy.

Reviewer:

**The authors should make clear that the amplitude scattering matrix (equation 11) operates on the complex electric field terms.**

[Figure]

Answer:

We added that the scattering amplitude matrix is a complex matrix and that it operates on the electric field, whereas the extinction, the scattering, and the Mueller matrix operate on the Stokes vector, which is a real vector.

Reviewer:

**The authors should explicitly state that orientational averaging must be done incoherently, that is at the the Mueller (or Phase) matrix stage, due to the power terms in the top left block of the Phase matrix.**

Answer:

We added to Eq. 2 and 3 a statement that we assume independent scattering and that therefore we assume incoherent scattering.

Reviewer:

**When discussing mirror partners and mirror symmetry, please cite van de Hulst (1957) and Mishchenko (2002). There is a really nice explanation with stick figures in both publications.**

Answer:

Done as suggested.

Reviewer:

**Also in reference to mirror particles, for the RT simulations in section 6.1, were the particles averaged with their mirror partners (with respect to the incidence plane)? This is important for properly conditioning the Z12 and Z21 Phase matrix parameters for the target medium of preferential alignment with zero mean canting angle.**

Answer:

No averaging of the scattering data of the particles with its mirrored version was done for the radiative transfer simulation. Due to the orientational averaging and the random structure of the large plate aggregate the effect of the non-mirror symmetry are so small, that we neglected it for the radiative transfer simulations. Particles like the plate type 1 atomatically fulfill this, as they are mirrorsymmetric. We added a similar statement to the text adressing this.

Technical corrections:

Reviewer:

**There are minor typos throughout the manuscript that need to be fixed, but the document as a whole is very clear.**

Answer:

Corrected them.

Reviewer:

**There are a few symbols that did not render properly, one of which was the asymmetry parameter.**

Answer:

Unfortunately, there were some problems with the font. This happened when the manuscript was uploaded to AMT. We are aware of this.

---

## Author Comment (AC3) · 31 Jan 2020

**Answers to Interactive comment on "Microwave andsubmillimeter wave scattering of oriented ice particles" by Manfred Brath et al.**

Davide Ori (dori@uni-koeln.de)

January 31, 2020

Reviewer:

**The paper presents an important step forward in the currently available scattering databases of snow particles at microwave frequencies by assuming the possibility of ice particles with preferential orientations. This is an important contribution which I recommend for publication, but I would also like to list some comments aiming to improve the value of the paper.**

**1. The orientation averaging technique lacks some validation. A very basic sanity check would be to calculate the integral over $\cos(\beta)$ at the various $\theta_{inc}$ and compare with the previously published database (DB) for total random orientation (TRO). Another useful plot to include would be the convergence of the integral with respect to the number of points of the icosahedral grid. At line 195 it is stated that a variable number of points is used (between 162 and 2562), per-**

**haps these convergence plots would clarify why, sometimes, a smaller number of orientation samples is sufficient.**

Answer:

You are right this is missing. We forgot to mention it in the text. We tested our method by simulating the scattering of azimuthally randomly oriented prolate ellipsoids and compared the results against T-matrix calculations. The overall differences in view of the extinction matrix and the scattering matrix were in the order of a few percent. We added a similar statement to the text. Considering line 195, we revised it. We now explain, why sometimes, a smaller number of orientations are sufficient.

Reviewer:

**2. The averaging scheme is presented as a solution to various challenges that sequentially appear in the text. It is hard, sometimes, to follow this approach because it requires to rethink about the setup many times without a clear final goal to aim to. I want to suggest to introduce the three main reference frames of the problem from the beginning: these are the laboratory (satellite) reference frame, the particle reference frame, and the wave reference frame. By doing so, one can states from the beginning that the scope is to have the polarized scattering properties defined with respect to the satellite reference frame and some transformations are needed because for scattering calculations the wave reference frame is a more natural option used in scattering codes. Also what is called the orientation of the non-rotated particle is nothing less than the particle reference frame.**

Answer:

We agree that it is sometimes hard to follow. We revised that part considering on your suggestion.

Reviewer:

**3. Line 62. This phrase, somehow implies that there is a special subset of rotation matrices that are orthogonal and no couple of rotation matrices are commutative with respect to multiplication. I think all rotation matrices are orthogonal and some rotation matrices do commute (the ones around the same axis).**

Answer:

We rephrased the sentence to: " It is important to note that in general the order of the rotations must not be changed, because the combination of rotations is generally not commutative."

Reviewer:

**4. Line 87. For TRO $p(\beta)$ should be $\frac{1}{2}$ and $\beta$ should be uniformly distributed in terms of $\cos(\beta)$. Otherwise, the integral does not compute to 1 when K=1**

[Figure]

Answer:

We corrected it. We now define $p(\beta)$ according to Mishchenko and Yurkin (2017) Eq. 4. This means

$$p(\beta) = \frac{\sin\beta}{2}\,. \tag{1}$$

Due to that we adjusted all equations in the text that involve averaging over the tilt angle $\beta$ (Eq. 2,3, and 37).

Reviewer:

**5. Line 110. I think here the non-symmetry is respect to the scattered azimuth, not the incident which is actually irrelevant for Zaro.**

Answer:

We removed "to incidence azimuth direction" from that specific sentence.

Reviewer:

**6. Line 121. ADDA can actually also compute scattering properties for distributions of angles through input files, this includes azimuthally averaging. The reason why this is not used in the study is that this approach involves the solution of the computationally demanding DDA problem for slightly different orientations many times (for the different combinations of tilt angle and wave incidence).**

[Figure]

Answer:

We rephrased it. We now state that the internal averaging method of ADDA is not suitable for our approach.

Reviewer:

**7. Line 130.** $D_0$ **should have explicit units, which I assume are** $\mu m$**.**

Answer:

We added the unit.

Reviewer:

**8. Line 179-182. I do not see why a regular grid is advantageous for resolving the for/back-ward scattering peaks. A regular grid means that the azimuth and polar angles are equally spaced. The points at the same polar angle are getting closer in azimuth distance as the polar angle approaches the poles. The scattering peaks mean that there is a high variability of the scattering intensity with respect to the polar angle and thus would demand an increased resolution in polar angles. The polar angle resolution is always the same here.**

Answer:

We removed that sentence.

Reviewer:

**9. Lines 209-214. In my opinion, two points are missing in the list of steps: first is the projection over spherical harmonics of the scattered fields. And the second is the barycentric interpolation of the gridded data. The second is important because it clarifies that the computed properties for a certain $\beta$ and $\theta_i$ are actually coming from slightly different angles.**

Answer:

We agree that the projection on spherical harmonics was missing in that list especially due to the truncation of them to reduce the amount of data. Therfore, we added it. We do not think that the barycentric interpolation is missing, because we think it is part of the averaging operation as the Gauss-Legendre quadrature is part of the averaging operation. Furthermore, the interpolation is explicitly stated in the paragraph before the list of steps. Therefore, we did not add it to the list.

Reviewer:

**10. Line 220. The three rotation matrices are different. Perhaps a better notation would be $R_{\alpha\beta\gamma} = R_\alpha\left(\alpha\right)R_\beta\left(\beta\right)R_\gamma\left(\gamma\right)$**

Answer:

Changed as suggested

Reviewer:

**11. Line 284. What is called accuracy $\epsilon = 1\%$ I think is the internal stopping criterion for the ADDA iterative solver and should not be confused with the accuracy of the calculations which is hard to evaluate and yet not clearly understood. Perhaps the authors should include in the supplementary material, for just one particle and one orientation what is the effect on the scattering properties (just plot phase functions ) of this choice of $\epsilon$ with respect to the default value of $10^{-5}$ (three orders of magnitude smaller!).**

Answer:

Your are right. We revised that part. We now say explicitly that it is the internal stopping criterion. Furthermore, we state that we aim for an accuracy of a few percent for our database. Therefore, setting the stopping criterion to $10^{-2}$ is a compromise in terms of accuracy due to the high demands in view of computation time and the amount of data. Considering the measurement errors of existing and upcoming passive MW and SubMM sensors, which are in the order of $\mathcal{O}(1\,K)$, and the brightness temperature depression due to scattering of frozen hydrometeors, which is typically $< 100\,K$, an accuracy of the scattering database in the order of a few percent seems sufficient.

Reviewer:

**12. Line 381. In the figure, I see $\beta$= 0, 50, 90 but in the text, $\beta$=30 is mentioned, perhaps there is a typo?**

Answer:

Yes, that were typos.

Reviewer:

**13. Line 397-402. Here the authors state that the database is not optimized for radar calculations because the spherical harmonics projection is not good at forward and backward scattering. Perhaps the authors should better describe what they meant at line 177 with RMSE of less than 0.5% due to the spherical harmonics. 0.5% is actually quite insignificant for radar applications. Also this problem can be immediately solved by making available the original DDA computations at single orientations, perhaps by request to the corresponding author. I think this last piece would also make the paper fully compliant with the Copernicus open-data policy.**

Answer:

Considering line 177, we revised that. We now relate the truncation of the spherical harmonics in Section Scattering calculations to the desired accuracy. We have to admit, that due to the missing statements on the desired accuracy of the database, it was not clear why we used an RMSE of 0.5%.

We cannot make the original DDA computations available. We could not store them permanently, because the data was too big. But we can make the truncated data from DDA computations available upon request. We added a statement considering the data availability to the text.

Reviewer:

**14. The scattering properties of hexagonal crystals are symmetric with respect to $\theta_i$ due to the planar symmetry of the particles. This is not true for aggregates that are not symmetric. The authors have oriented the aggregates according to their principal axis of inertia. This is, in general, a good fast approach, but it introduces an arbitrary decision about what is the direction of the main (vertical) axis of inertia. In my opinion, there is no clear criterion to decide whether this axis should look up or down. As a consequence, one could argue that the scattering properties for $\theta_i = \lambda$ should be averaged with those for $\theta_i = 180 - \lambda$ giving planar symmetry also to the aggregates and reducing the storage footprint of the database.**

Answer:

You are correct. When using the axis of inertia there is no unique criterion to decide whether the axis should be upward or downward. Therefore, we use an additional criterion, see Appendix Initial particle alignment step 4. We define, if the center of the circumscribed sphere of the particle is found to be below the mass-center of the particle (with respect to the z-axis), then the particle is said to be aligned upright and vice versa. We did not consider your suggested averaging, because we want the users to decide what they want or need.

Reviewer:

**15. Equations (20) and (21) show how to rotate the polarization vectors of the Mueller matrix. I wonder if this is done before the barycentric interpolation. In my view, the scattering properties of the three vertexes should be first aligned**

**with the direction and polarization of point D. If the forward/backward scattering direction lies within the triangle ABC this can cause quite dramatic cancellation due to the flipping of the polarization direction among the points A, B, and C.**

Answer:

We do not fully understand your point. Equations (20) and (21) describe how the Mueller matrix and the extinction matrix are transformed. The interpolation is done with respect to the incidence direction not with respect to the scattering directions. Actually, the points (vertices) of the triangle are the three nearest sample points to our desired incidence direction at point D after the rotation of the particle system. The sample points are the set of the incidence directions at which we have calculated the Mueller matrix with ADDA. At each of the three nearest sample points we transform the polarization according to Eq. 22 using the Stokes matrices. And after that we do the interpolation, which is essentially a weighted averaged of the Mueller matrices of these three sample points.

**0.1  Refrences**

Mishchenko, M. I. and Yurkin, M. A.: On the concept of random orientation in far-field electromagnetic scattering by nonspherical particles, Optics letters, 42, 494-497, 2017.